# On Reinforcement Learning and Distribution Matching for Fine-Tuning Language Models with no Catastrophic Forgetting

**Tomasz Korbak**[*]
University of Sussex
tomasz.korbak@gmail.com

**Hady Elsahar**
Naver Labs Europe
hady.elsahar@gmail.com

**Germán Kruszewski**
Naver Labs Europe
german.kruszewski@naverlabs.com

**Marc Dymetman**[†]
marc.dymetman@gmail.com

## Abstract

The availability of large pre-trained models is changing the landscape of Machine Learning research and practice, moving from a "training from scratch" to a "fine-tuning" paradigm. While in some applications the goal is to "nudge" the pre-trained distribution towards preferred outputs, in others it is to steer it towards a different distribution over the sample space. Two main paradigms have emerged to tackle this challenge: Reward Maximization (RM) and, more recently, Distribution Matching (DM). RM applies standard Reinforcement Learning (RL) techniques, such as Policy Gradients, to gradually increase the reward signal. DM prescribes to first make explicit the target distribution that the model is fine-tuned to approximate. Here we explore the theoretical connections between the two paradigms, and show that methods such as KL-control developed for RM can also be construed as belonging to DM. We further observe that while DM differs from RM, it can suffer from similar training difficulties, such as high gradient variance. We leverage connections between the two paradigms to import the concept of *baseline* into DM methods. We empirically validate the benefits of adding a baseline on an array of controllable language generation tasks such as constraining topic, sentiment, and gender distributions in texts sampled from a language model. We observe superior performance in terms of constraint satisfaction, stability and sample efficiency.

## 1 Introduction

Pre-trained language models (Devlin et al., 2019; Radford et al., 2019) are changing the landscape of Machine Learning research and practice. Due to their strong generative capabilities many studies have found it sufficient to "nudge" these models to conform to global preferences defined over the generated sequences instead of training from scratch using annotated data. These preferences could include topic and sentiment (Dathathri et al., 2020), valid musical notes and molecular structures (Jaques et al., 2017a), code compilability (Korbak et al., 2021), reducing distributional biases (Khalifa et al., 2021; Weidinger et al., 2021), evaluation metrics for Machine Translation and Summarization (Ranzato et al., 2016; Bahdanau et al., 2016), or direct human feedback (Ziegler et al., 2019; Stiennon et al., 2020). This large body of studies is driven by two different paradigms: *Reward Maximization* (RM) and *Distribution Matching* (DM).

---

[*]Work partly done during an internship at Naver Labs Europe.
[†]Independent Researcher. Work done at Naver Labs Europe.

36th Conference on Neural Information Processing Systems (NeurIPS 2022).

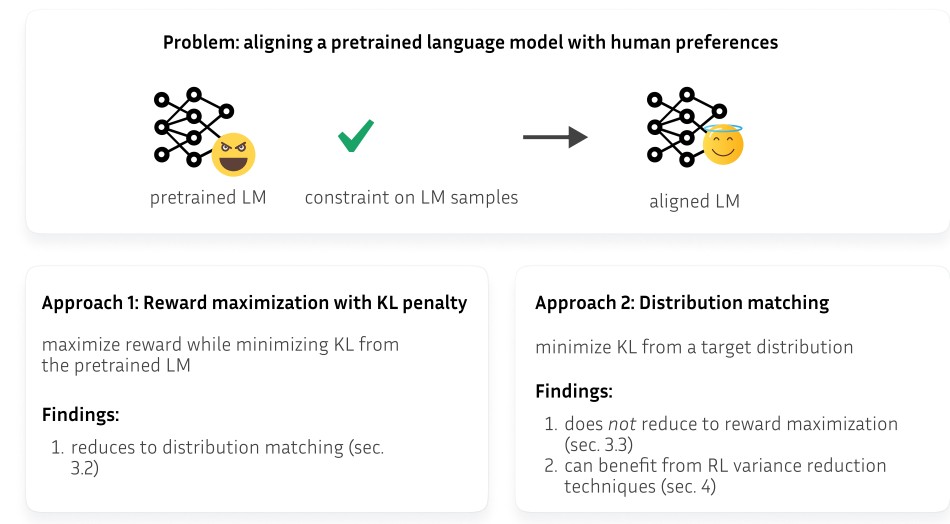

Figure 1: In this study we make a connection between two popular paradigms for aligning language models to human preferences: Reward maximization (RM) and Distribution matching (DM).

**Reward Maximization** RM intuitively nudges pre-trained models towards certain preferences by providing global sequence-level rewards when the model generates outputs that satisfy desired features. For instance, if the model is producing toxic content, we can apply Reinforcement Learning (RL) techniques to discourage it from producing similar content. However, naively applying RL yields a model that can undergo *catastrophic forgetting* of its original distribution. For example, it can degenerate into producing a single nonsensical but at least nontoxic sequence. Although several studies have considered hand-crafting general rewards to ensure desirable features like fluency (Liu et al., 2016a; Tambwekar et al., 2019), coming up with complete or perfect rewards is highly non-trivial (Wu et al., 2016; Vedantam et al., 2015). This has sparked a wide discussion on the overall effectiveness of RM for some tasks such as machine translation (Choshen et al., 2020; Kiegeland & Kreutzer, 2021).

**Reward Maximization with KL-Control** To tackle the aforementioned issues of "catastrophic forgetting", several studies, still under an RM paradigm, have considered incorporating a distributional term inside the reward to be maximized. In particular Jaques et al. (2017b, 2019) and Ziegler et al. (2019) or more recently Stiennon et al. (2020), Ouyang et al. (2022), Bai et al. (2022) and Perez et al. (2022) have applied variations of KL-control (Todorov, 2007; Kappen et al., 2012) which adds a penalty term to the reward term so that the resulting policy does not deviate too much from the original one in terms of KL-divergence. The overall objective with the KL-penalty is maximized using an RL algorithm of choice including: PPO (Schulman et al., 2017a) as in Ziegler et al. (2019) or Bai et al. (2022) or Q-learning (Mnih et al., 2013) as in Jaques et al. (2017b). Adding this *distributional* KL-penalty to the reward raises some important questions: What effect does it have on the shape of the optimal policy? Does this new objective have any interpretation from a distributional perspective?

**Distribution Matching** A different recent paradigm for fine-tuning language models to satisfy downstream preferences formulates the problem as Distribution Matching (DM). This paradigm consists of two steps: first a target distribution incorporating the desired preferences is defined as an Energy-Based Model (LeCun et al., 2006). Then the forward KL divergence is minimized between this target distribution and an auto-regressive policy using a family of algorithms referred to as Distributional Policy Gradients (DPG) (Parshakova et al., 2019b; Khalifa et al., 2021; Korbak et al., 2021, 2022a). This approach capitalizes on the flexibility of EBMs in specifying the target distribution. For example, the EBM can be defined so that it conforms to all downstream preferences while its corresponding normalized distribution has a minimal KL divergence from the original, pre-trained language model, therefore tackling the problem of "catastrophic forgetting" (Khalifa et al., 2021). Interestingly, this DM paradigm can also deal with *distributional* preferences, for instance, for de-biasing language models by specifying that the generated sequences should be gender-balanced,

i.e. that 50% of generations contain female mentions. Such distributional constraints cannot be defined in the RM paradigm where a reward is calculated for a single sequence.

We can notice the promises and limitations of these two paradigms for fine-tuning language models. RM approaches are equipped with an arsenal of RL algorithms and optimization techniques that can be efficient in reward maximization, however they lack the distributional aspect to avoid catastrophic forgetting and impose distributional preferences over LMs. DM approaches are suited to tackle those limitations, however, the family of DPG algorithms currently used is not as rich as its RL counterpart.

While the connections between these two seemingly distinct paradigms have been noted (Parshakova et al., 2019b; Korbak et al., 2022b), they have not been explored in detail. Clarifying such connections might help import ideas from one approach to the other. This is our goal in this paper, detailing the nuanced connections and applying them to a case-study in variance reduction. Overall, our contributions are the following:

- We clarify relations between the RM and DM paradigms through a detailed comparison between the family of DPG algorithms and Policy Gradients (Table 1), stressing the differences between *parametric* and *non-parametric* rewards that are important in this regard.

- We introduce an interpretation of KL-control techniques from a distribution matching perspective, placing such techniques at an intermediate place between RM and DM (Theorem 1).

- We show how these connections can enable cross-pollination between the two perspectives by applying *baselines* — a variance reduction technique from RL — to DPG and derive a particular choice of a baseline (Facts 1 and 2). On an array of controllable language generation experiments, we show that adding baselines leads to superior performance on constraint satisfaction (Figure 3), stability on small batch sizes, and sample efficiency (Figure 4).

## 2   Background

**Standard Policy Gradients**   One popular method for adapting the behaviour of language models to certain preferences has been that of assigning a "reward" score $R(x)$ for sequences $x$ sampled from an autoregressive language model (policy) $\pi_\theta$. Then, the simplest policy gradient algorithm in reinforcement learning, namely, REINFORCE (Williams, 1992a), aims to find the policy $\pi_\theta(x)$ that maximizes the average reward $\mathbb{E}_{x \sim \pi_\theta} R(x)$, and this leads, via the so-called "log derivative trick", to a gradient ascent algorithm that iteratively samples $x$ from $\pi_\theta$ and update parameters by increments proportional to $R(x)\nabla_\theta \log \pi_\theta(x)$ via the following identity:

$$\nabla_\theta \mathbb{E}_{x \sim \pi_\theta} R(x) = \mathbb{E}_{x \sim \pi_\theta} R(x)\nabla_\theta \log \pi_\theta(x). \tag{1}$$

**KL-control**   (Todorov, 2007; Kappen et al., 2012), was leveraged by Jaques et al. (2017b, 2019) and Ziegler et al. (2019) to include a KL penalty term in the reward function to penalize large deviations from the original pretrained model $a(x)$, weighted by a free hyperparameter $\beta$ to control the trade-off between the two goals. That is, they maximize the expectation $\mathbb{E}_{x \sim \pi_\theta} R_\theta^z(x)$, where:

$$R_\theta^z(x) \doteq r(x) - \beta \log \frac{\pi_\theta(x)}{a(x)}. \tag{2}$$

**Distributional Policy Gradients**   (DPG) (Parshakova et al., 2019b) is a recent approach used to fit an autoregressive policy $\pi_\theta$ to the distribution $p(x) = P(x)/Z$ induced by the EBM $P(x)$, where $Z = \sum_x P(x)$ is the normalization constant (partition function). Given an arbitrary EBM $P(x)$, DPG optimizes the loss function $D_{\mathrm{KL}}(p, \pi_\theta)$ with respect to the parameters $\theta$ of an autoregressive model $\pi_\theta$, a loss which is minimized for $\pi_\theta = p$. The KL-divergence minimization objective leads to a gradient estimate of the form:

$$\nabla_\theta D_{\mathrm{KL}}(p, \pi_\theta) = -\nabla_\theta \mathbb{E}_{x \sim p} \log \pi_\theta(x) \tag{3}$$

$$= -\sum_x p(x)\nabla_\theta \log \pi_\theta(x) = -\frac{1}{Z} \sum_x P(x)\nabla_\theta \log \pi_\theta(x) \tag{4}$$

$$= -\frac{1}{Z} \mathbb{E}_{x \sim \pi_\theta} \frac{P(x)}{\pi_\theta(x)} \nabla_\theta \log \pi_\theta(x). \tag{5}$$

# 3 Reward Maximization vs Distribution Matching

In the previous section, we have summarized three approaches that have been suggested for fine-tuning language models. Two of them can be characterized as "Reward Maximization" (RM): Standard Policy Gradients (PG) and KL-control. On the other hand, DPG clearly belongs to the realm of "Distribution Matching" (DM) as it first defines the target distribution and then optimizes a policy to match it. In the rest of this section, we will explore connections between these two seemingly distinct concepts and, in the following section, we will exploit them to improve DM-based methods.

## 3.1 Standard vs. Parametric Rewards

Let us start with distinguishing between a "parametric reward" $R_\theta$ which depends on $\theta$ and a standard reward $R$, which does not. If we wished to maximize the expected parametric reward, $\mathbb{E}_{\pi_\theta} R_\theta(x)$, we would follow its gradient, leading to the identities:

$$\nabla_\theta \mathbb{E}_{x \sim \pi_\theta} R_\theta(x) = \nabla_\theta \sum_x \pi_\theta(x) R_\theta(x) = \sum_x \pi_\theta(x) \nabla_\theta R_\theta(x) + \sum_x R_\theta(x) \nabla_\theta \pi_\theta(x) \quad (6)$$

$$= \sum_x \pi_\theta(x) \nabla_\theta R_\theta(x) + \sum_x \pi_\theta(x) R_\theta(x) \nabla_\theta \log \pi_\theta(x) \quad (7)$$

$$= \underbrace{\mathbb{E}_{x \sim \pi_\theta} \nabla_\theta R_\theta(x)}_{\text{RG-term}} + \underbrace{\mathbb{E}_{x \sim \pi_\theta} R_\theta(x) \nabla_\theta \log \pi_\theta(x)}_{\text{PG-term}}. \quad (8)$$

Equation (8) is the sum of two terms: the first one, the "RG-term" (Reward Gradient term), involves the gradient of the reward. The second one, the "PG-term" (Policy Gradient term), was obtained using the "log derivative trick" and involves the gradient of the policy *stricto sensu*. In standard RL, where the reward does *not* depend on $\theta$, the RG-term disappears and the gradient of expected reward consists solely of the PG-term. However, when $R_\theta$ depends on $\theta$, the gradients are distinct (apart from specific cases where the RG-term evaluates to 0, as we will see below).

## 3.2 KL-control as Distribution Matching

Adding a KL-penalty term to the reward (as in the case of KL-control) leads to a parametric reward. However, due to the particular form of its objective, the RG-term actually *vanishes*,[3] leaving only the PG-term $\mathbb{E}_{x \sim \pi_\theta} R_\theta^z(x) \nabla_\theta \log \pi_\theta(x)$ and simplifying the tuning procedure to a standard Policy Gradient. While this algorithm falls under the RM paradigm, here we argue that is its nature is multifaceted, and explore deeper connections with the DM paradigm. More precisely, the maximization of reward with the KL penalty term is equivalent to a distributional matching with an underlying emergent sequential EBM, a remark that already reveals some similarities with DPG.[4]

**Theorem 1.** *Consider the following EBM:*

$$P_z(x) = a(x) e^{r(x)/\beta} \quad (9)$$

*and let $p_z$ be the normalized distribution $p_z(x) = \frac{1}{Z} P_z(x)$, with $Z = \sum_x P_z(x)$. Then:*

*(i)* $\arg\max_{\pi_\theta} \mathbb{E}_{x \sim \pi_\theta} R_\theta^z(x) = \arg\min_{\pi_\theta} D_{\text{KL}}(\pi_\theta, p_z)$;

*(ii)* $\arg\max_{\pi \in \mathcal{D}(X)} \mathbb{E}_{x \sim \pi} R_\pi^z(x) = p_z$, *where $\mathcal{D}(X)$ is the family of all distributions over $X$, and $R_\pi^z(x) \doteq r(x) - \beta \log \frac{\pi(x)}{a(x)}$.*

*Proof.* A simple way to prove this is to notice that the expectation of the reward $R_\theta^z$ has a monotonically decreasing relationship with the *reverse* KL divergence between $\pi_\theta$ and $p_z$:

$$D_{\text{KL}}(\pi_\theta, p_z) = \mathbb{E}_{x \sim \pi_\theta} \log \frac{\pi_\theta(x)}{p_z(x)} = \mathbb{E}_{x \sim \pi_\theta} \left[ \log \pi_\theta(x) - \log \frac{1}{Z} a(x) e^{r(x)/\beta} \right]$$

---

[3]This is because $\mathbb{E}_{\pi_\theta} \nabla_\theta R_\theta^z(x) = -\beta \, \mathbb{E}_{\pi_\theta} \nabla_\theta \log \pi_\theta(x) = 0$, via the identity $\mathbb{E}_{\pi_\theta} \nabla_\theta \log \pi_\theta(x) = \sum_x \pi_\theta(x) \nabla_\theta \log \pi_\theta(x) = \sum_x \nabla_\theta \pi_\theta(x) = \nabla_\theta \sum_x \pi_\theta(x) = 0$.

[4]The optimal policy $p_z$ is briefly mentioned in (Ziegler et al., 2019) without reference or derivation. The proof, which reveals a connection to the reverse KL divergence from $\pi_\theta$, is ours.

| | Policy Gradients | DPG |
|---|---|---|
| **Reward** | $R(x)$ | $R_\theta(x) = \frac{P(x)}{\pi_\theta(x)}$ |
| $\nabla_\theta$ | $\mathbb{E}_{x \sim \pi_\theta} R(x) \nabla_\theta \log \pi_\theta(x)$ | $\mathbb{E}_{x \sim \pi_\theta} \frac{P(x)}{\pi_\theta(x)} \nabla_\theta \log \pi_\theta(x)$ |
| **Baseline** | $\mathbb{E}_{x \sim \pi_\theta} R(x)$ | $Z$ |
| $\nabla_\theta$ **with Baseline** | $\mathbb{E}_{x \sim \pi_\theta} \left[ R(x) - \mathbb{E}_{x \sim \pi_\theta} R(x) \right] \nabla_\theta \log \pi_\theta(x)$ | $\mathbb{E}_{x \sim \pi_\theta} \left[ \frac{P(x)}{\pi_\theta(x)} - Z \right] \nabla_\theta \log \pi_\theta(x)$ |

Table 1: A comparison between Policy Gradients (Sutton et al., 1999) and Distributional Policy Gradients (Parshakova et al., 2019b) forms of Reward, Baseline, and Gradient of the loss function (the PG-term) before ($\nabla_\theta$) and after ($\nabla_\theta$ with Baseline) including a baseline for variance reduction .

$$= \log Z - \frac{1}{\beta} \mathbb{E}_{x \sim \pi_\theta} \left[ r(x) - \beta \log \frac{\pi_\theta(x)}{a(x)} \right] = \log Z - \frac{1}{\beta} \mathbb{E}_{x \sim \pi_\theta} R_\theta^z(x), \qquad (10)$$

so that the $\arg\min_{\pi_\theta} D_{\mathrm{KL}}(\pi_\theta, p_z)$ coincides with the $\arg\max_{\pi_\theta} \mathbb{E}_{x \sim \pi_\theta} R_\theta^z(x)$, proving (i). On the other hand, $\arg\min_{\pi \in \mathcal{D}(X)} D_{\mathrm{KL}}(\pi, p_z)$, which also corresponds to $\arg\max_{\pi \in \mathcal{D}(X)} \mathbb{E}_{x \sim \pi} R_\pi^z$ because of (i) applied to a family $\pi_{\theta'}$ covering $\mathcal{D}(X)$ in full, is just $p_z$, concluding the proof. $\qquad \square$

Overall, we can conclude that the addition of the distributional term (KL-penalty) to the reward does indeed provide a DM interpretation, namely in terms of minimizing the reverse KL divergence with an emergent underlying distribution $p_z(x)$. We note that $p_z(x)$ does not correspond to a free and explicit choice of EBM (e.g. one that balances the gender and topic distributions of a language model). Instead equation (9) appears in a restrictive format, which is implicitly defined by the reward $R_\theta^z$, along with a $\beta$ hyperparameter without a clear meaning. By contrast, the DPG algorithms are designed to perform DM on any EBM specification, corresponding to an explicit distributional objective.

### 3.3 Similarities and Differences between DPG and Policy Gradients

In the previous subsection, we have connected KL-control, a method designed under a RM paradigm, to DM. Now, we turn to the converse question of whether DPG, a DM method, can be connected to RM. We begin by noting that after defining $R_\theta = \frac{P(x)}{\pi_\theta(x)}$, the DPG gradient $\mathbb{E}_{x \sim \pi_\theta} \frac{P(x)}{\pi_\theta(x)} \nabla_\theta \log \pi_\theta(x)$ acquires the format of the PG-term $\mathbb{E}_{\pi_\theta} R_\theta \nabla_\theta \log \pi_\theta(x)$.

However, the DM objective of DPG *cannot* be considered as maximizing the average "reward" $R_\theta(x) = \frac{P(x)}{\pi_\theta(x)}$, as this would require adding also the RG-term $\mathbb{E}_{\pi_\theta} \nabla_\theta \frac{P(x)}{\pi_\theta(x)}$ into the gradient, which in this case does not vanish.

Nonetheless, the analogy behind this gradient term is more fruitful than it first appears. As a matter of fact, DPG gradient estimates suffer from the same high-variance problems as with standard PG. While the objective of DPG (distribution matching) is different from that of Policy Gradients (reward maximization), DPG also needs to estimate the PG-term $\mathbb{E}_{\pi_\theta} R_\theta(x) \nabla_\theta \log \pi_\theta(x)$ at a *given* value of $\theta$, using a batch of samples $x$. For such a *fixed* $\theta$, we can define provisionally set $R(x) \doteq R_\theta$ and the problem of gradient estimation *for this fixed $\theta$* is identical to the estimation $\mathbb{E}_{x \sim \pi_\theta} R(x) \nabla_\theta \log \pi_\theta(x)$ based on a set of samples $x$ in standard RL. Therefore, techniques that have been developed to reduce the variance of the gradients estimates in RL can be ported to DPG insofar as we are computing the gradient estimates *at a given $\theta$*. In Section 4, we show how one can import one such variance reduction technique to the DPG: baselines.

## 4 A Case Study on Variance Reduction

Baselines are a standard variance reduction technique in the context of Policy Gradients (Sutton & Barto, 2018). The idea is to subtract from the reward $R(x)$ a value $B$ that does not introduce bias to the gradients but may change variance. After the introduction of baseline, equation (1) then takes the following form:

$$\nabla_\theta \mathbb{E}_{\pi_\theta} R(x) = \mathbb{E}_{\pi_\theta} [R(x) - B] \nabla_\theta \log \pi_\theta(x). \qquad (11)$$

In standard RL, the simplest form of baseline $B$ is just the average of the rewards for the policy:[5]

$$B^{\text{RL}} = \mathbb{E}_{x \sim \pi_\theta} R(x). \tag{12}$$

Following the same methodology of taking the baseline to be the expectation of the reward term, we can obtain a remarkably simple form of a baseline for DPG:[6]

$$B = \mathbb{E}_{x \sim \pi_\theta} \frac{P(x)}{\pi_\theta(x)} = \sum_x \pi_\theta(x) \frac{P(x)}{\pi_\theta(x)} = \sum_x P(x) = Z. \tag{13}$$

**Fact 1.** *Subtracting $B$ from $R_\theta(x)$ does not introduce bias into DPG gradient estimates.*

*Proof.* Let us rewrite the DPG gradient in (5) with the added baseline $B = Z$:

$$\mathbb{E}_{x \sim \pi_\theta} \Big[ R_\theta(x) - Z \Big] \nabla_\theta \log \pi_\theta(x) = \mathbb{E}_{x \sim \pi_\theta} R_\theta(x) \nabla_\theta \log \pi_\theta(x) - Z \, \mathbb{E}_{x \sim \pi_\theta} \nabla_\theta \log \pi_\theta(x)$$

$$= \mathbb{E}_{x \sim \pi_\theta} R_\theta(x) \nabla_\theta \log \pi_\theta(x) - Z \Big[ \sum_x \nabla_\theta \pi_\theta(x) \Big] \tag{14}$$

Here, the second term does not introduce bias because $Z \Big[ \sum_x \nabla_\theta \pi_\theta(x) \Big] = 0$, leaving us with the exact same form of gradient as in the original DPG algorithm. □

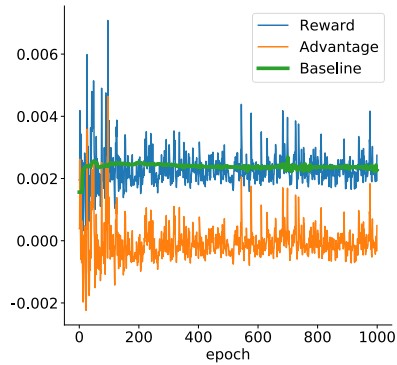

Figure 2: Values of reward, advantage and the baseline for first 1000 epochs of a point-wise constraint experiment.

Note that since $B^{\text{RL}}$ depends on $\theta$, it has to be be re-estimated after each gradient update. On the other hand, $B$ does *not* depend on $\theta$, which is an advantage because $B$ could be now estimated by averaging over samples from *all* the different $\theta$'s without introducing bias, leading to a more accurate estimation. See Table 1 for a comparison of these two forms of baselines.

The off-policy DPG version introduced in (Parshakova et al., 2019b) and its KL-adaptive variant (Khalifa et al., 2021) sample a proposal distribution $q$ instead of the policy $\pi_\theta$. Then, the baseline takes the form

---

**Algorithm 1** KL-Adaptive DPG with baseline

---

**Require:** $P$, initial generative model $a$
1:  $\pi_\theta \leftarrow a, q \leftarrow a$
2: **for** each iteration **do**
3:     **for** each episode **do**
4:         sample $x$ from $q(\cdot)$
5:         $\theta \leftarrow \theta + \alpha^{(\theta)} \Big[ \frac{P(x)}{q(x)} - Z \frac{\pi_\theta(x)}{q(x)} \Big] \nabla_\theta \log \pi_\theta(x)$
6:         **if** $D_{\text{KL}}(p||\pi_\theta) < D_{\text{KL}}(p||q)$ **then**
7:             $q \leftarrow \pi_\theta$
**Ensure:** $\pi_\theta$

---

$$B^{\text{off}}(x) = Z \frac{\pi_\theta(x)}{q(x)}, \tag{15}$$

where the $\frac{\pi_\theta(x)}{q(x)}$ term is an importance weight correcting for the bias introduced by sampling from $q$. Similarly to the DPG case, we can prove the following (see Appendix C):

**Fact 2.** *Subtracting $B^{off}(x)$ from $R_\theta(x)$ does not bias the off-policy DPG gradient estimates.*

In practice, as shown on Figure 2, adding a baseline to KL-adaptive DPG (Algorithm 1) centers the advantage (defined as $A \doteq \frac{P(x)}{q(x)} - Z \frac{\pi_\theta(x)}{q(x)}$) around 0 leading to better performance on: convergence (section 4.3), stability on small batch sizes (section 4.4), and variance reduction (section 4.5).

---

[5]While this baseline is not optimal (proof Appendix C.1), it is widely used in practice.

[6]In the scope of this paper, our focus is on importing to DPG simple constant baselines. The advantage is that this is a technique that is not impacted by the fact that $R_\theta$ depends on $\theta$: it can be applied "$\theta$-locally" to provide a more accurate estimate of $\mathbb{E}_{x \sim \pi_\theta} R_\theta(x) \nabla_\theta \log \pi_\theta(x)$ for a *fixed* $\theta$, irrespective of the values of $R_{\theta'}$ elsewhere, while variance reduction techniques that involve several $\theta's$ simultaneously raise additional challenges for parametric rewards.

## 4.1 Generation with Distributional Control

We investigate the benefits of adding a baseline to the DPG algorithm, on the Generation with Distributional Control (GDC) (Khalifa et al., 2021) framework. GDC makes use of DPG to control the properties of pre-trained language models to satisfy certain constraints. In our experiments, follow target distribution form of Parshakova et al. (2019a), Khalifa et al. (2021) and Korbak et al. (2022a), in which the EBM $P(x)$ is defined so that its normalized variant $p(x)$ matches a set of desired moments constraints on given features $\phi_i(x)$, while having a minimal KL divergence $D_{\mathrm{KL}}(p, a)$ from an original pretrained language model $a$, to avoid catastrophic forgetting.

These constraints are expressed as conditions $\bar{\mu}_i = \mathbb{E}_{x \sim p} \phi_i(x)$, for $i \in \{1, \ldots, n\}$, by which the moments (expectations) under the distribution $p$ of each feature $\phi_i(x)$ are required to take certain desired values $\bar{\mu}_i$. For instance, let $\phi_1(x) = 1$ iff the topic of $x$ is science and $\phi_2(x) = 1$ iff $x$ mentions a female person, then imposing moments $\bar{\mu}_1 = 1$ and $\bar{\mu}_2 = 0.5$ constrains the language model $p$ to only generate sequences about science, half of which mention females. $P(x)$ is uniquely determined by the following form:[7]

$$P(x) = a(x) e^{\sum_{i=1}^{n} \lambda_i \phi_i(x)}, \tag{16}$$

where $\lambda_i$ terms control the moments $\mu_i$ of the associated features, which can be estimated through self-normalized importance sampling (Owen, 2013); and then, to make the moments match the desired values, the $\lambda_i$ terms can be optimized through SGD (Parshakova et al., 2019a).

## 4.2 Experimental setup

We evaluate our method on an array of 10 controlled text generation tasks. For each, given a pre-trained language model $a(x)$, and a set of constraints, the objective of each fine-tuning method is to obtain a fine-tuned language model $\pi_\theta$ that satisfies the imposed constraints while deviating as minimally as possible from the original language model $a(x)$.

Constraints are defined as a set of binary features $\{\phi_i\}$ and their corresponding desired percentages (moments) $\{\bar{\mu}_i\}$ within the generations of the target language model. Based on the value of the moment constraints these 10 tasks are divided into 6 tasks of pointwise constraints (for which $\bar{\mu}_i = 1$), 2 tasks of distributional constraints ($0 < \bar{\mu}_i < 1$) and 2 tasks of mixed type constraints (hybrid):

(a) Single-word constraints, where $\phi(x) = 1$ iff the a given word appears in the sequence $x$. We experiment with frequent words (task 1: "amazing", original frequency: $10^{-4}$) and (task 2: "WikiLeaks", original frequency: $10^{-5}$) rare words,

(b) Wordlist constraints, where $\phi(x) = 1$ iff $x$ contains at least one word from a given list. We consider lists of word associated with politics (task 3) and science (task 4) published by Dathathri et al. (2020),

(c) Sentiment classifier constraints, where $\phi(x) = 1$ if $x$ is classified as positive (task 5), or negative (task 6) by a pre-trained classifier published by Dathathri et al. (2020).

(d) A single distributional constraint where $\phi(x) = 1$ iff $x$ contains a female figure mention, and $\bar{\mu} = 0.5$ (task 8),

(e) A set of four distributional constraints: $\phi_i(x) = 1$ iff $x$ contains at least one of the words in the "science", "art", "sports" and "business" wordlists (compiled by Dathathri et al. (2020)), respectively. For each $i$, $\bar{\mu}_i = 0.25$ (task 8),

(f) Hybrid constraints where $\phi_1(x) = 1$ iff $x$ contains more female than male pronouns, $\bar{\mu}_1 = 0.5$ and $\phi_2(x) = 1$ iff $x$ contains at least one of the words from the "sports" wordlist (task 9) or "politics" wordlist, $\bar{\mu}_2(x) = 1$ (task 10).

**Methods** We modify the GDC framework Khalifa et al. (2021), namely its KL-DPG algorithm, to include a baseline as shown in Algorithm 1. We refer to this method as **GDC++**. In addition to comparing **GDC++** with **GDC** we compare with two reward maximization baselines: **Reinforce** (Williams, 1992b) and **Ziegler** (Ziegler et al., 2019). Reinforce tries to maximize the expected reward $\mathbb{E}_{x \sim \pi_\theta} R(x)$, where $R(x) = 1$ if and only if the pointwise constraints are met. Ziegler instantiates the KL-control approach: its objective includes a KL penalty term for departures from $a$. Following (Khalifa et al., 2021), for hybrid and distributional constraints (tasks 8-10) we compare

---

[7]For a more precise formulation of this EBM, see (Khalifa et al., 2021).

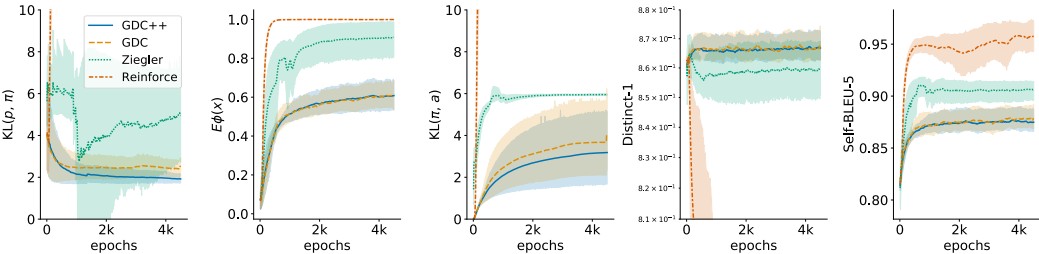

Figure 3: Evaluation metrics: $D_{\mathrm{KL}}(p, \pi_\theta)$ ($\downarrow$ better), $\mathbb{E}_{\pi_\theta} \phi(x)$ ($\uparrow$ better), $D_{\mathrm{KL}}(\pi_\theta, a)$ ($\downarrow$ better), Self-BLEU-5 ($\downarrow$ better), and Distinct-1 ($\uparrow$ better) aggregated over 6 pointwise constraints experiments (tasks 1-6) for policies obtained from GDC++, GDC, Ziegler and Reinforce. See Figure 6 for aggregated distributional constraints experiments. In the Appendix Figures 7-10 and Table 4 contain individual view and final results of each run.

only GDC and GDC++ because the RM objective of Ziegler and Reinforce is not equipped to handle them.

**Metrics** We report the following metrics at each validation step over batches of samples from $\pi_\theta$:

1. $\mathbb{E}_{x \sim \pi_\theta} \phi_i(x)$, measuring the ability to reach the target moment of the $i$-th feature.
2. $D_{\mathrm{KL}}(p, \pi_\theta)$, the forward KL divergence from the optimal target distribution $p$.[8]
3. $D_{\mathrm{KL}}(\pi_\theta, a)$, the reverse KL divergence from the original pretrained language model $a$.
4. Distinct-n score, a measure of text diversity in terms of the frequency of repetitions within a single sample $x$, proposed by (Li et al., 2016a).
5. Self-BLEU-n, a measure of text diversity on a distributional level *across* samples proposed by (Zhu et al., 2018), ensuring that policies don't converge into limited number of sequences that satisfy the imposed constraints Caccia et al. (2020).

**Training details** For tasks 1-6, we use a pre-trained GPT-2 small with 117M parameters (Radford et al., 2019) as the original language model $a$. For tasks 7-10, $a$ is the same pre-trained model additionally fine-tuned on the WikiBio (Lebret et al., 2016) dataset. See Appendix E for more details. The code for all the experiments presented in the paper will be available at github.com/naver/gdc.

## 4.3 Results

We present the evolution of our metrics through training epochs in Figure 3 (aggregated over tasks 1-6) and Figure 6 in the Appendix (aggregated over tasks 7-10). Results for each task are presented separately on Figures 7-10 in the Appendix.

Consistent with prior work (Khalifa et al., 2021; Korbak et al., 2022a), we observe that Reinforce is able to quickly achieve high levels of constraint satisfaction, but at the cost of large deviations from $a$, which translates into significantly decreased diversity of generated samples (in terms of Self-BLEU-5 and Distinct-1). The KL penalty term in Ziegler imposes an upper bound on deviation from $a$ but the deviation is still significant enough to result in a drop in diversity. Moreover, we have observed Ziegler's objective to result in very unstable training.

GDC and GDC++ are the only fine-tuning methods that address constraint satisfaction based on a clear formal objective, i.e. reducing the divergence from $p$. The approach translates into significantly smaller deviations from $a$ and maintaining diversity within and across samples. The addition of a baseline indeed reduces the variance. We analyze that extensively in Appendix 4.5 while here focusing on the downstream effects of variance reduction. One is that $\pi_\theta$ is now able to compound staying closer to $p$ and $a$ *at the same time*, while achieving slightly better constraint satisfaction. We have also observed that baseline stabilizes training, leading to smoother curves.[9]

## 4.4 The effect of baseline across batch sizes

We expect that reducing gradient estimates variance can allow to train with lower batch sizes, performing gradient updates on estimates based on smaller batch sizes can increase the sample

---

[8]See Appendix D for a detailed description of how $D_{\mathrm{KL}}(p, \pi_\theta)$ is computed.

[9]The interested reader can compare the large fluctuations of the Ziegler objective to more stable training curves of GDC , and even more of GDC++ , in the disaggregated curves in Figures 7-10 of the Appendix.

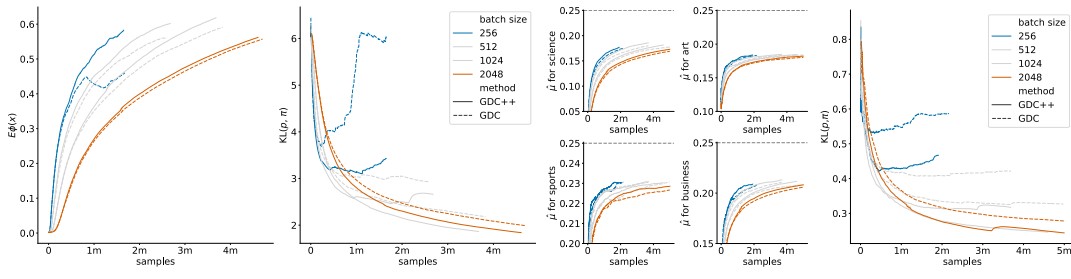

(a) Task 1: a pointwise constraint

(b) Task 8: a set of distributional constraints; $\bar{\mu}_i = 0.25$

Figure 4: $\mathbb{E}_{\pi_\theta}\phi(x)$ or $\hat{\mu}$ per constraint ($\uparrow$ better) and $D_{\mathrm{KL}}(p, \pi_\theta)$ ($\downarrow$ better) as a function of the number of samples reported for task 1 (a) and task 8 (b). We report the number of samples (i.e. the number of epochs times the batch size) for a fair comparison of convergence speed. *GDC++ is consistently superior across all batch sizes in terms of convergence and constraint satisfaction.* The effect is more conspicuous with small batch sizes. Batch sizes 512 and 2014 are greyed out for clarity.

efficiency. To test this, we rerun tasks 1 (a pointwise constraint on the word "amazing") and 8 ( distributional constraints on topics) with four batch sizes (256, 512, 1024, 2048). Figures 4a and 4b show the benefits of adding a baseline — higher constraint satisfaction, lower divergence from $p$, more stable training — and is especially evident with lower batch sizes. For instance, with batch size 256, GDC++ obtains a significantly higher constraint satisfaction rate and lower divergence from $p$.

Furthermore, stable training with smaller batch sizes translates into better sample efficiency. For instance, in task 1 (Figure 4a), GDC++ with batch size 256 needs 1M samples to achieve $\mathbb{E}_{x\sim\pi_\theta}\phi(x) = 0.5$ while GDC++ with batch size 2048 needs 4M. In contrast, GDC with batch size 256 does not achieve $\mathbb{E}_{x\sim\pi_\theta}\phi(x) = 0.5$ at all, confirming the importance of adding the baseline.

### 4.5 Empirical Evaluation of Variance Reduction

Next, we evaluate empirically the effect of the baseline for variance reduction. We select two tasks: task 1 (a pointwise constraint) and task 7 (distributional constraints) described in Section 4.2, each with 3 different seeds, while monitoring the following variance measures:

**Gradient Variance** The gradient estimate is defined as: $G_\theta(x) \doteq A(x)\nabla_\theta \log \pi_\theta(x)$, where $G_\theta(x) \in \mathbb{R}^{|\theta|}$ is an unbiased estimate of the gradient of the forward KL loss $\nabla_\theta D_{\mathrm{KL}}(p, \pi_\theta)$ with respect to the parameters $\theta$. We then have, with $\mu(G_\theta) \doteq \mathbb{E}_{x\sim q}G_\theta(x)$:

$$\mathrm{Var}(G_\theta) \doteq \mathbb{E}_{x\sim q} \|G_\theta(x) - \mu(G_\theta)\|_2^2 \quad (17)$$

$$= \mathbb{E}_{x\sim q}||G_\theta(x)||_2^2 - ||\mu(G_\theta)||_2^2. \quad (18)$$

**Variance of the advantage** is defined by:

$$\mathrm{Var}(A) \doteq \mathbb{E}_{x\sim q} \left\| A(x) - \mu^A \right\|_2^2 \quad (19)$$

where, $\mu^A \equiv \mathbb{E}_{x\sim q} A(x)$ is the mean of the advantage, which we showed above to be null after the addition of the baseline.

**Expected absolute value of the advantage** This metric is defined as:

$$\mu^{|A|} \doteq \mathbb{E}_{x\sim q} |A(x)|. \quad (20)$$

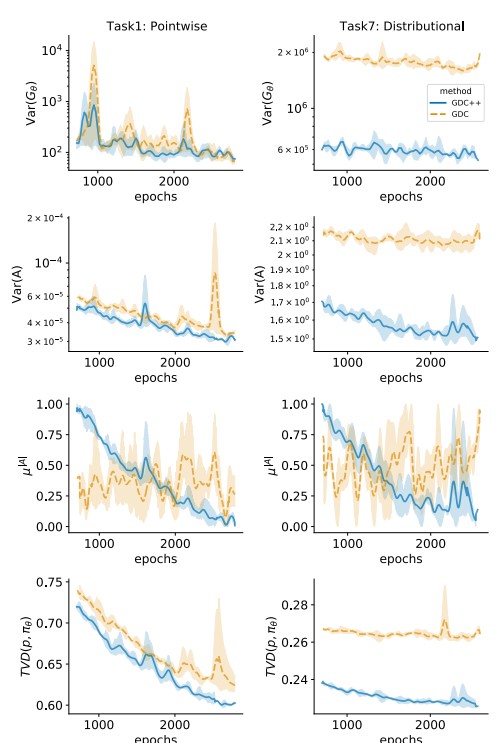

Figure 5: Comparison between GDC and GDC++ using a set of Variance diagnosis metrics on pointwise and distributional constraints experiments.

It directly provides a standard measure of distributional discrepancy between $p$ and $\pi_\theta$, in terms of TVD (Total Variation Distance). We have:

$$\mathbb{E}_{x \sim q} \left| \frac{p(x)}{q(x)} - \frac{\pi_\theta(x)}{q(x)} \right| = 2\,\mathrm{TVD}(p, \pi_\theta). \tag{21}$$

**Results**   Figure 5 shows that GDC++ obtains lower variance in the gradient estimates $\mathrm{Var}(G_\theta)$ and the variance of the advantage $\mathrm{Var}\,(A)$ in both pointwise and distributional experiments compared to its non-baseline counterpart GDC. We further observe a decreasing trend in the mean absolute value of the advantage $\mu^{|A|}$ which is correlated with a decreasing trend in the TVD distance between the trained policy $\pi_\theta$ and the optimal distribution $p$. Overall, these results shows that adding a baseline to DPG reduces the variance during training and yields better convergence towards the optimal distribution $p$.

## 5   Related work

The idea of posing control problems as distribution matching has resurfaced numerous times in the RL literature (Kappen et al., 2012; Friston et al., 2010; Levine, 2018; Hafner et al., 2020; Buckley et al., 2017). KL-control can be seen as a generalisation of maximum entropy RL (MaxEnt RL) (Haarnoja et al., 2017, 2018) to informed priors. If in (2) we chose $a(x)$ to be a uniform distribution (assuming right now finiteness of $X$) instead of a pretrained LM distribution, then the KL penalty $D_{\mathrm{KL}}(\pi_\theta, a)$ would reduce to an entropy bonus. Both KL-control and MaxEnt RL can be derived from a general framework of control-as-inference (Levine, 2018) which poses control as minimising KL from a certain target distribution. However, most practical algorithms in the MaxEnt RL family minimise KL from a target policy which changes throughout training; in contrast, DPG's target distribution $p$ and KL-control implicit target distribution $p_z$ are defined at trajectory level and fixed throughout training.

Perhaps the closest method to KL-control and DPG in the larger family of inference-based RL (Furuta et al., 2021) is AWR (Peng et al., 2019) which minimises the *forward* KL from an off-policy target distribution. Yet another approach with apparent similarity to KL-control and DPG is state marginal matching (SMM) (Hazan et al., 2018; Lee et al., 2019). SMM poses exploration as learning a policy that induces a state marginal distribution that matches a target state distribution. While SMM's target distribution is fixed, it is defined for individual states, while in the controllable language generation tasks we consider, the target distribution is defined over a complete trajectory considered as a unit. See Appendix B for an extended discussion of related work.

## 6   Conclusion

Fine-tuning large language models has become an active area of research, due to its importance in adapting large language models to satisfy task-level preferences, or in combating their social risks such as "distributional" stereotyping (Weidinger et al., 2021; Welbl et al., 2021). [10] In this paper, we analyzed in depth the nuanced relation between two popular fine-tuning paradigms: RM and DM. We demonstrated that KL-control can be seen as a form of DM and showed that while DPG and PG have different goals, some similarities (similar forms of gradient estimates despite different objectives) can be exploited. We used these insights to inform an extension of DPG, consisting in adding a baseline to reduce the variance of gradient estimates.

The connections we established suggest that despite fundamental differences between DPG and RL, some of the theoretical results and algorithmic techniques from RL can be adapted to a DM framework without losing their formal guarantees. In this paper, we focus on variance reduction using baselines, but the space of possible enhancements is vast. Promising candidates include further reducing the variance using a learned value function (Konda & Tsitsiklis, 2000) and preventing detrimentally large policy updates by maintaining a trust region in the policy space – akin to techniques such as TRPO (Schulman et al., 2015) and PPO (Schulman et al., 2017b). Another future direction could consist in analyzing the relation between explicit EBMs in DPG and implicit EBMs arising in KL-control and characterizing the space of EBMs that could be reached through KL-control.

---

[10] See Appendix A for a discussion of broader impacts of large language models and controllable language generation.

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
