# A    Broader impacts

The focus area of this paper — fine-tuning large language models — is aligned with an important line of work on addressing the problem of social bias in large language models (Sheng et al., 2019; Liang et al., 2021). As the training data for large language models consists mainly of crawled user-generated content, a number of factors (from crawling methodology to Internet participation inequalities and moderation practices) leads to an over-representation of certain viewpoints and voices exceeding their prevalence in the general population. This poses a risk of amplifying biases and harms through a language model perpetuating these voices (Bender et al., 2021; Blodgett et al., 2020; Sheng et al., 2019; Weidinger et al., 2021; Welbl et al., 2021). Numerous problems related to addressing data bias in language generation (e.g. controlling for gender distribution in generated texts) can be naturally posed as generative distributional control (GDC), the framework we focus our experiments on. The *distributional* character of these data bias problems lies in the fact that desirable properties of generated texts are defined for a collection of samples, not only for individual samples. Our theoretical analyses of reward maximization and distribution matching approaches as well as our algorithmic improvements to the GDC framework — termed GDC++ — are therefore also a contribution to the problem of bias in language models. However, we need to be aware that GDC++ , KL-control as well as controllable language generation techniques in general, can also be diverted to malicious uses such as spreading misinformation or generating harmful content.

# B    Extended Related Work

**Reinforcement learning for language generation**    Most previous attempts at steering language models to conform to global constraints defined over entire sequences have employed reinforcement learning. This includes using Reinforce (Williams, 1992a) for machine translation Ranzato et al. (2016), actor critic (Konda & Tsitsiklis, 2000) for abstractive summarization (Paulus et al., 2018), caption generation (Liu et al., 2016b), dialogue (Li et al., 2016b), and video captioning (Pasunuru & Bansal, 2017). Some approaches (for instance, in machine translation and summarization (Ranzato et al., 2016; Bahdanau et al., 2017)) directly optimize performance metrics such as BLEU and ROUGE at training time. Others use heuristic rewards (for instance Li et al. (2016b) for dialogue generation and Tambwekar et al. (2019) for story generation) in order to obtain certain *a priori* desirable features of generated sequences that then incentivize good performance on target metrics. Catastrophic forgetting is a frequent problem of these fine-tuning approaches: reward maximization happens at the expense of large deviations from the original model. This problem is sometimes addressed by imposing a penalty term to the rewards, such as the KL divergence between the trained policy and the auto-regressive model. This approach, termed "conservative fine-tuning", was applied to generating melodies with music theory rewards and organic molecules with synthesizability rewards by Jaques et al. (2017a) as well fine-tuning language models for controllable language generation by Ziegler et al. (2019). This solution often has hard time balancing between the reward term and the KL penalty term, leading to instability in training  (Khalifa et al., 2021; Korbak et al., 2022a). Unlike this approach, KL-DPG determines an optimal distribution that satisfies both requirements.

**RM and DM objectives in control problems**    While RM is the dominant approach to tackling control problems (Sutton & Barto, 2018) and is sometimes argued to be sufficient for any intelligent behavior (Silver et al., 2021), prior work explored the benefits of alternative objectives formulated as DM: minimizing divergence from some target distribution $p$. Prominent examples of (families of) DM objectives include control state marginal matching (Lee et al., 2019) active inference (Friston et al., 2010; Buckley et al., 2017) and control-as-inference (Kappen et al., 2012; Todorov, 2007; Levine, 2018). Hafner et al. (2020) propose a *reverse* KL from a joint distribution over observations and latent variables as a universal objective for action and perception that — depending on a choice of the target $p$ — gives rise to many familiar objectives, including empowerment (Klyubin et al., 2005), maximum entropy RL (Haarnoja et al., 2017) or KL-control (Todorov, 2007). In a similar vein, Millidge et al. (2021) compare RM and DM objectives (or, evidence and divergence objectives, according to their terminology) in the context of exploration. They conclude that information-seeking exploration arises naturally in DM but *not* in RM. This is because, when the target distribution $p$ involves latent variables, a DM objective decomposes into an information gain term that pushes the agent to seek observations that are most informative of latent variables. In contrast, RM objectives entail *minimizing* information gain between latent variables and observations. Finally, (Korbak et al.,

2022b) defend an interpretation of KL-control for controlling language models as Bayesian inference: updating a prior $a$ to conform to evidence provided by a reward function $R$.

**Maximum entropy RL**    Maximum entropy RL (MaxEnt RL)'s objective is maximising expected reward minus policy entropy. KL-control can be seen as generalisation of maximum-entropy RL (Haarnoja et al., 2017, 2018) to informed priors. If in (2) we chose $a(x)$ to be a uniform distribution (an uninformed prior) instead of a pretrained LM distribution, then the KL penalty $D_{\mathrm{KL}}(\pi_\theta, a)$ would reduce to an entropy bonus and KL-control's objective would reduce to a standard Maximum entropy RL objective. Both KL-control and Maximum entropy RL can be derived from a general framework of control-as-inference (Levine, 2018) which poses control as minimising KL from a certain target distribution. However, while KL-control (Ziegler et al., 2019) and DPG directly minimise a single KL from a target distribution over whole sequences (trajectories), most practical algorithms in the maximum entropy family RL approximate it by related but importantly different objectives.

The three biggest differences between MaxEnt RL on the one hand and DPG and KL-control (Ziegler et al., 2019) on the other hand are as follows:

1. KL-control implicit target distribution $p_z$ and DPG's target distribution $p$ are over whole sequences (trajectories) while in most MaxEnt RL algorithms the target distribution over actions conditioned on a state: $\pi^*(a|s)$. For instance in both SQL (Haarnoja et al., 2017) and SAC (Haarnoja et al., 2018) the target distribution is defined as $\pi^*(a|s) = \exp(Q_\theta(s,a))/Z_\theta(s)$, where $Q$ is a state-action value function and $Z$ is a partition function of for a given state, both dependent on policy parameters $\theta$.

2. KL-control's implicit target distribution and DPG's target distribution are predefined (i.e. held constant throughout training). In MaxEnt RL it typically undergoes updates. Again, in both SQL (Haarnoja et al., 2017) and SAC (Haarnoja et al., 2018) they depend on a Q function which is continuously updated on new trajectories.

3. KL-control's implicit target distribution $p_z$ and DPG's target distribution $p$ involve an informed prior $a(x)$: a pretrained language model. In most MaxEnt RL algorithms, the prior is assumed to be a uniform distribution.

Because MaxEnt RL algorithms do not approximate a constant, predefined target distribution, they cannot be framed as minimising a single KL objective. Instead, they typically implement (soft) policy iteration (Sutton & Barto, 2018): they alternate between defining a new target distribution (policy evaluation) and minimising KL from that current target distribution (policy improvement). In other words, minimising KL is a subroutine of policy iteration, not an objective in itself.

Perhaps the closest method to KL-control and DPG in the larger family of inference-based RL (Furuta et al., 2021) is AWR (Peng et al., 2019), which minimises the *forward* KL from a target distribution $\frac{1}{Z}\mu(a|s)\exp(A(s,a))$, where $\mu$ is a behavioural policy implicitly defined by the trajectory buffer and $A$ is the advantage. Here, the prior is informative and given by the policy from a previous iteration $k$. However, the target distribution is not constant: it is updated on each iteration.

**State marginal matching**    State marginal matching (Hazan et al., 2018; Lee et al., 2019) is an approach to exploration in RL. It poses exploration as learning a policy $\pi$ that induces a state marginal distribution $\rho_\pi(s) = \mathbb{E}\sum_{t=1}^{T} \mathbb{1}(s_t = t)$ that matches a given target state distribution $p^*$. While this approach differs in motivation from DPG and KL-control (it solves the problem of exploration in the space of policies, not constraint satisfaction), it optimises a similar divergence objective: $D_{\mathrm{KL}}(\pi, p^*)$. Unlike in maximum-entropy RL, the target $p^*$ is fixed. However, $p^*$ is a distribution over states, not trajectories (as in the case of $p$ in DPG and $p_z$ in KL-control). There is no obvious notion of state in the controllable language generation tasks we consider other than treating the whole sequence as a state.

**Baselines in Reinforcement Learning**    In the context of reinforcement learning, baselines were introduced by Sutton (1984). Williams (1987, 1992a) has shown them to reduce variance in a number of use cases and also proved that they do not introduce bias. Dayan (1990) was the first to observe and confirm experimentally that the optimal constant baseline is not equal to expected reward in a simple two-arm bandit setting. This result was generalized to POMDPs (Partially Observable Markov Decision Processes) by Weaver & Tao (2001, section 3.1.3, p. 540) and variable baselines by

Greensmith et al. (2004, theorem 13, p. 1489) who also proved bounds on the variance of gradient estimates. The optimal baseline, however, is rarely used in practice (Sutton & Barto (2018); for an exception, see (Peters & Schaal, 2008)). Outside RL, baselines were also used in the context of learning inference networks for amortized variational inference by Mnih & Gregor (2014) and found to yield similar variance reduction.

**Energy-based models for language** Energy-based models (EBMs) (Hinton, 2002; LeCun et al., 2006; Ranzato et al., 2007) are a family of models in which learning and inference are done by associating an unnormalized probability with each configuration of observed and latent variables. Early examples of EBMs applied to natural language processing include sequence labeling problems (e.g. tagging) exploiting global properties of a sequence (Andor et al., 2016; Belanger & McCallum, 2016). The recent surge of interest in EBMs has not left natural language processing unaffected (see Bakhtin et al. (2020) for a survey). Tu et al. (2020) proposed an energy-based inference networks for non-autoregressive machine translation while Naskar et al. (2020) use an EBM for reranking candidate translations according to their predicted BLEU scores. Parshakova et al. (2019a) and Deng et al. (2020) augment an autoregressive language models with an additional global factor to obtain a lower perplexity on the training data. Clark et al. (2020) poses non-autoregressive language modeling as training an energy-based cloze task scorer using noise-contrastive estimation (Gutmann & Hyvärinen, 2010). He et al. (2021) obtain better calibration on natural language inference tasks by augmenting and training the classifier jointly with an energy-based model modeling the marginal distribution over samples, again using noise-contrastive estimation. In consequence, the classifier tends to assign more conservative (high-entropy) predictions to high-energy (less likely, possibly out of distribution) samples.

# C  Additional proofs

## C.1  Optimal baselines in RL

Despite its widespread use, the baseline as mean of reward

$$B^{\text{RL}} = \mathbb{E}_{x \sim \pi_\theta(x)} R(x) \tag{22}$$

is not the optimal constant baseline for reward maximization objectives in RL. The optimal constant baseline, i.e. one yielding the minimal variance of the gradient, is given by:

$$B^* = \frac{\mathbb{E}_{x \sim \pi_\theta} \left[ R(x) \left( \nabla_\theta \log \pi_\theta(x) \right)^2 \right]}{\mathbb{E}_{x \sim \pi_\theta} \left[ \left( \nabla_\theta \log \pi_\theta(x) \right)^2 \right]}. \tag{23}$$

In order to maintain accessibility, in this section, we provide a self-contained derivation of this optimal form of baselines (23) and and connect it to the commonly used form (22).[11]

First, recall that $R(x)$ is a reward associated with an input $x$. $B$ is a baseline value subtracted from the reward that does not introduce bias in gradient estimation. Now let's denote the gradient wrt an individual sample $x$ as $G_\theta(x)$ where

$$G_\theta(x) = [R(x) - B] \nabla_\theta \log \pi_\theta(x), \tag{24}$$

and the estimate of the gradient as

$$G(\theta) = \mathbb{E}_{x \sim \pi_\theta} G_\theta(x). \tag{25}$$

Using the general identity $\mathbf{var}(z) = \mathbb{E}[z^2] - [\mathbb{E}z]^2$, the variance of the gradient takes the form:

$$\text{Var}(G_\theta) = \mathbb{E}_{x \sim \pi_\theta} [G_\theta(x)^2] - G(\theta)^2 \tag{26}$$

Now let's take the gradient of this variance with respect to $B$ and solve to find the baseline form with minimal variance:

$$\frac{d\text{Var}(G_\theta)}{dB} = \frac{d}{dB} \mathbb{E}_{x \sim \pi_\theta} [(G_\theta(x))^2] - \frac{d}{dB} (\mathbb{E}_{x \sim \pi_\theta} [G_\theta(x)])^2. \tag{27}$$

---

[11]The formula for the optimal baseline in (23) was originally proved by Weaver & Tao (2001) but here we provide a simpler proof sketched by Sergey Levine in his slides: `http://rail.eecs.berkeley.edu/deeprlcourse-fa17/f17docs/lecture_4_policy_gradient.pdf`

The second term of the right hand side of (27) is equal to zero, since $B$ does not introduce bias into $G(\theta)$:

$$\frac{d}{dB}\left(\mathbb{E}_{x\sim\pi_\theta}[G_\theta(x)]\right)^2 = \frac{d}{dB}\left(\mathbb{E}_{x\sim\pi_\theta}\left[(R(x)-B)\nabla_\theta\log\pi_\theta(x)\right]\right)^2$$
$$= \frac{d}{dB}\left(\mathbb{E}_{x\sim\pi_\theta}\left[R(x)\nabla\log\pi_\theta(x)\right]\right)^2 = 0.$$

Plugging this back into (27), we obtain:

$$\frac{d\mathrm{Var}(G_\theta)}{dB} = \frac{d}{dB}\mathbb{E}_{x\sim\pi_\theta}[(G_\theta(x))^2]$$
$$= \mathbb{E}_{x\sim\pi_\theta}\left[\frac{d}{dB}\left[\left(R(x)^2+B^2-2R(x)B\right)(\nabla_\theta\log\pi_\theta(x))^2\right]\right]$$
$$= \mathbb{E}_{x\sim\pi_\theta}(2B-2R(x))(\nabla_\theta\log\pi_\theta(x))^2$$
$$= 2B\,\mathbb{E}_{x\sim\pi_\theta}(\nabla_\theta\log\pi_\theta(x))^2 - 2\,\mathbb{E}_{x\sim\pi_\theta}R(x)\left(\nabla_\theta\log\pi_\theta(x)\right)^2.$$

Then, solving $\frac{d\mathrm{Var}(G_\theta)}{dB}=0$ for $B$, we obtain the optimal form of the baseline $B^*$ as required:

$$B^* = \frac{\mathbb{E}_{x\sim\pi_\theta}[R(x)\left(\nabla_\theta\log\pi_\theta(x)\right)^2]}{\mathbb{E}_{x\sim\pi_\theta}[\left(\nabla_\theta\log\pi_\theta(x)\right)^2]}. \tag{28}$$

This can be interpreted as average reward (as in $B^{\mathrm{RL}}$) but weighted by gradient magnitudes $(\nabla_\theta\log\pi_\theta(x))^2$. Moreover, $B^* = B^{\mathrm{RL}}$ is recovered **under the condition that** the reward $R(x)$ is uncorrelated (*a fortiori* independent) from $(\nabla_\theta\log\pi_\theta(x))^2$. If that were the case, we would have:

$$B^* = \frac{\mathbb{E}_{x\sim\pi_\theta}[R(x)\left(\nabla_\theta\log\pi_\theta(x)\right)^2]}{\mathbb{E}_{x\sim\pi_\theta}[\left(\nabla_\theta\log\pi_\theta(x)\right)^2]} \tag{29}$$

$$= \frac{\mathbb{E}_{x\sim\pi_\theta}[R(x)]\,\mathbb{E}_{x\sim\pi_\theta}[\left(\nabla_\theta\log\pi_\theta(x)\right)^2]}{\mathbb{E}_{x\sim\pi_\theta}[\left(\nabla_\theta\log\pi_\theta(x)\right)^2]} \tag{30}$$

$$= \mathbb{E}_{x\sim\pi_\theta}[R(x)] = B^{\mathrm{RL}}. \tag{31}$$

### C.2 unbiasedness of PG baseline

Baselines are a standard variance reduction technique in the context of Policy Gradients (Sutton & Barto, 2018). The idea is to subtract from the reward $R(x)$ a value $B$ that does not introduce bias to the gradients but may change variance. Equation (1) then takes the following form:

$$\nabla_\theta\mathbb{E}_{\pi_\theta}R(x) = \mathbb{E}_{\pi_\theta}(R(x)-B)\,\nabla_\theta\log\pi_\theta(x). \tag{32}$$

To see that $B$ does not introduce bias, we can rewrite (11) as:

$$\mathbb{E}_{x\sim\pi_\theta}R(x)\nabla_\theta\log\pi_\theta(x) - B\,\mathbb{E}_{\pi_\theta}\nabla_\theta\log\pi_\theta(x) \tag{33}$$

and note that the second term is null because $\sum_x\pi_\theta(x)\nabla_\theta\log\pi_\theta(x) = \nabla_\theta\sum_x\pi_\theta(x) = 0$.

### C.3 Unbiasedness of DPG Baseline

Recall that the gradient estimate for DPG (Parshakova et al., 2019a) has the following form:

$$\mathbb{E}_{x\sim\pi_\theta}\frac{P(x)}{\pi_\theta(x)}\nabla_\theta\log\pi_\theta(x) \tag{34}$$

After subtracting a baseline $B=Z$, it becomes

$$\mathbb{E}_{x\sim\pi_\theta}\left[\frac{P(x)}{\pi_\theta(x)}-Z\right]\nabla_\theta\log\pi_\theta(x) = \mathbb{E}_{x\sim\pi_\theta}\frac{P(x)}{\pi_\theta(x)}\nabla_\theta\log\pi_\theta(x) - Z\left[\mathbb{E}_{x\sim\pi_\theta}\nabla_\theta\log\pi_\theta(x)\right] \tag{35}$$

$$= \mathbb{E}_{x\sim\pi_\theta}\frac{P(x)}{\pi_\theta(x)}\nabla_\theta\log\pi_\theta(x) - Z\left[\sum_x\nabla_\theta\pi_\theta(x)\right] \tag{36}$$

Here, the second term does not introduce bias because $Z\left[\sum_x\nabla_\theta\pi_\theta(x)\right]=0$, leaving us with the same exact form of gradient as in the DPG algorithm.

## C.4 Unbiasedness of DPGoff baseline

Offline DPG, the off policy variant of DPG proposed in Parshakova et al. (2019b); Khalifa et al. (2021) has the following gradient estimate:

$$\mathbb{E}_{x \sim q} \frac{P(x)}{q(x)} \nabla_\theta \log \pi_\theta(x) \tag{37}$$

Where $q$ is a proposal distribution (another auto-regressive model) used to detach the training of $\pi_\theta$ from the sampling process and allow more stable training.

Recall that the Baseline of DPGoff is of the form:

$$B^{\text{off}}(x) = Z \frac{\pi_\theta(x)}{q(x)}, \tag{38}$$

The $\frac{\pi_\theta(x)}{q(x)}$ term is an importance weight correcting for the bias introduced by sampling from $q$.

**Unbiasedness**   To show that subtracting a baseline $B^{\text{off}}(x) = Z \frac{\pi_\theta(x)}{q(x)}$ doesn't introduce bias, let's rewrite the gradient estimate with added baseline as a sum of two terms:

$$\mathbb{E}_{x \sim q} \Big[ \frac{P(x)}{q(x)} - Z \frac{\pi_\theta(x)}{q(x)} \Big] \nabla_\theta \log \pi_\theta(x) = \Big[ \mathbb{E}_{x \sim q} \frac{P(x)}{q(x)} \nabla_\theta \log \pi_\theta \Big] - \Big[ \mathbb{E}_{x \sim q} Z \frac{\pi_\theta(x)}{q(x)} \nabla_\theta \log \pi_\theta \Big] \tag{39}$$

$$= \Big[ \mathbb{E}_{x \sim q} \frac{P(x)}{q(x)} \nabla_\theta \log \pi_\theta \Big] - Z \Big[ \sum_x \nabla_\theta \pi_\theta(x) \Big] \tag{40}$$

Here again the second term does not introduce bias because $Z \Big[ \sum_x \nabla_\theta \pi_\theta(x) \Big] = 0$.

**Null Advantage on Average**   In the case of sampling with $\pi_\theta$ in the online DPG choosing $B = Z$ had the benefit that the advantage $R_\theta(x) - B$ was centered around 0, namely: $\mathbb{E}_{x \sim \pi_\theta}[R_\theta(x) - Z] = 0$.

With the $B^{\text{off}}(x)$ baseline for the DPGoff this important property is also maintained. The advantage now takes the form $\frac{P(x)}{q(x)} - Z \frac{\pi_\theta(x)}{q(x)}$ and then:

$$\mathbb{E}_{x \sim q} \Big[ \frac{P(x)}{q(x)} - Z \frac{\pi_\theta(x)}{q(x)} \Big] = \sum_x P(x) - Z \pi_\theta(x) \tag{41}$$

$$= Z - Z \sum_x \pi_\theta(x) = 0. \tag{42}$$

To visualize things better, we elaborate the difference in forms of rewards, baseline and gradients before and after addition of the baseline between DPG (on policy) and DPGoff (off policy) in Table 2.

|  | **DPG** | **DPG$^{\text{off}}$** |
|---|---|---|
| **Reward** | $\frac{P(x)}{\pi_\theta(x)}$ | $\frac{P(x)}{q(x)}$ |
| $\nabla_\theta$ | $\mathbb{E}_{x\sim\pi_\theta}\frac{P(x)}{\pi_\theta(x)}\nabla_\theta\log\pi_\theta(x)$ | $\mathbb{E}_{x\sim q}\frac{P(x)}{q(x)}\nabla_\theta\log\pi_\theta(x)$ |
| **Baseline** | $Z$ | $Z\frac{\pi_\theta(x)}{q(x)}$ |
| **Advantage** | $\frac{P(x)}{\pi_\theta(x)}-Z$ | $\frac{P(x)}{q(x)}-Z\frac{\pi_\theta(x)}{q(x)}$ |
| $\nabla_\theta$ **with baseline** | $\mathbb{E}_{x\sim\pi_\theta}\left[\frac{P(x)}{\pi_\theta(x)}-Z\right]\nabla_\theta\log\pi_\theta(x)$ | $\mathbb{E}_{x\sim q}\left[\frac{P(x)}{q(x)}-Z\frac{\pi_\theta(x)}{q(x)}\right]\nabla_\theta\log\pi_\theta(x)$ |

Table 2: A comparison of Online DPG and Offline DPG (DPG$^{\text{off}}$ ) forms of Reward, Baseline, Advantage, and Gradient of the loss function (the PG-term) before ($\nabla_\theta$) and after ($\nabla_\theta$ with Baseline) including a baseline for variance reduction.

## D   Additionals details on metrics and Algorithms

Calculation of metrics relative to $p$, such as $D_{\text{KL}}(p,\pi_\theta)$, is not straightforward since the distribution $p \propto P$ is only implicitly represented by the unnormalized EBM $P$, and one cannot easily obtain direct samples from $p$. Instead, we apply the following workarounds. Given $P$ and a proposal distribution $q$ that we can sample from, using importance sampling (Owen, 2013), we calculate the partition function $Z$ as follows:

$$Z = \sum_x P(x) = \sum_x q(x)\, P(x)/q(x) \tag{43}$$

$$= \mathbb{E}_{x\sim q}\, P(x)/q(x). \tag{44}$$

The precision of this estimate depends on the sample size and the quality of the proposal distribution $q$. We calculate a moving average estimate $Z_{\text{MA}}$ of $Z$ which is then used inside the estimations of $D_{\text{KL}}(p,\pi_\theta)$ and $D_{\text{KL}}(p,q)$ (see below Algorithm 2, lines 7 and 8). $Z_{\text{MA}}$ is updated at each training iteration. $Z_{\text{MA}}$ is an unbiased estimate of $Z$ because each $\hat{Z}_i$ is an unbiased estimate of $Z$ based on $K$ samples. Moreover, because the proposal distribution $q$ evolves and gets closer to the target distribution $p$, the quality of the estimate of $Z_{\text{MA}}$ through importance sampling increases.

With an estimate of $Z$, we can compute $D_{\text{KL}}(p,\pi_\theta)$ as

$$D_{\text{KL}}(p,\pi_\theta) = \sum_x p(x) \log \frac{p(x)}{\pi_\theta(x)} \tag{45}$$

$$= \sum_x p(x) \log \frac{P(x)}{Z\pi_\theta(x)} \tag{46}$$

$$= -\log Z + \sum_x p(x) \log \frac{P(x)}{\pi_\theta(x)} \tag{47}$$

$$= -\log Z + \sum_x q(x)\frac{p(x)}{q(x)} \log \frac{P(x)}{\pi_\theta(x)} \tag{48}$$

$$= -\log Z + \frac{1}{Z}\mathbb{E}_{x\sim q}\frac{P(x)}{q(x)} \log \frac{P(x)}{\pi_\theta(x)}. \tag{49}$$

Similarly, for $\text{TVD}(p,\pi_\theta)$:

$$\text{TVD}(p,\pi_\theta) = \frac{1}{2}\sum_x |p(x)-\pi_\theta(x)| \tag{50}$$

$$= \frac{1}{2}\sum_x q(x)\left|\frac{\pi_\theta(x)}{q(x)}-\frac{p(x)}{q(x)}\right| \tag{51}$$

$$= \frac{1}{2}\sum_x q(x)\left|\frac{\pi_\theta(x)}{q(x)}-\frac{P(x)}{Z\,q(x)}\right| \tag{52}$$

$$= \frac{1}{2} \mathbb{E}_{x \sim q} \left| \frac{\pi_\theta(x)}{q(x)} - \frac{P(x)}{Z \, q(x)} \right|. \tag{53}$$

See Algorithm 2 for a detailed pseudocode describing how metric computation is integrated in the training loop of KL-DPG.

---

**Algorithm 2** KL-DPG with baseline (detailed)

---

**Require:** $P$, initial policy $q$
1: $\pi_\theta \leftarrow q$
2: $Z_{\mathrm{MA}} \leftarrow 0$
3: **for** each iteration $i$ **do**
4:      **for** each step $k \in [1, K]$ **do**
5:          sample $x_k$ from $q(\cdot)$
6:          $\theta \leftarrow \theta + \alpha^{(\theta)} \left[ \frac{P(x_k)}{q(x_k)} - Z \frac{\pi_\theta(x_k)}{q(x_k)} \right] \nabla_\theta \log \pi_\theta(x_k)$
7:      $\hat{Z}_i \leftarrow \frac{1}{K} \sum_k P(x_k)/q(x_k)$
8:      $Z_{\mathrm{MA}} \leftarrow \frac{i * Z_{\mathrm{MA}} + \hat{Z}_i}{i+1}$
9:      $\hat{D}_{\mathrm{KL}}(p, \pi_\theta) \leftarrow -\log Z_{\mathrm{MA}} + 1/(K Z_{\mathrm{MA}}) \sum_k \frac{P(x_k)}{q(x_k)} \log \frac{P(x_k)}{\pi_\theta(x_k)}$
10:     $\hat{D}_{\mathrm{KL}}(p, q) \leftarrow -\log Z_{\mathrm{MA}} + 1/(K Z_{\mathrm{MA}}) \sum_k \frac{P(x_k)}{q(x_k)} \log \frac{P(x_k)}{q(x_k)}$
11:      **if** $\hat{D}_{\mathrm{KL}}(p, \pi_\theta) < \hat{D}_{\mathrm{KL}}(p, q)$ **then**
12:          $q \leftarrow \pi_\theta$
**Ensure:** $\pi_\theta$

---

# E   Hyperparameters and training details

We implemented all models using PyTorch (Paszke et al., 2019) and HuggingFace (Wolf et al., 2019). Based on Khalifa et al. (2021) source code published under CC BY-NC-SA 4.0 license: `https://github.com/naver/gdc`. The two pretrained models used in our experiments are available on Hugginface Model Hub: `gpt`[12] and `mkhalifa/gpt2-biographies`.[13] Each training run took approximately 5 days on 2 Nvidia V100 GPUs. For a detailed list of hyperparameter values, see Table 3; for a description of hyperparameters specific to Ziegler and GDC, see (Ziegler et al., 2019) and (Khalifa et al., 2021).

| Hyperparameter | Value |
|---|---|
| **Common** | |
| batch size | 512 |
| sequence length | 40 tokens |
| learning rate | $1.41 \times 10^{-5}$ |
| dropout rate | 0.1 |
| optimizer | Adam (Kingma & Ba, 2014) |
| warmup epochs | 100 |
| total epochs | 4500 |
| base LM | GPT-2 small (117M params) |
| **GDC** | |
| sample size for learning $\lambda$ | 10240 |
| learning rate for $\lambda$ | 0.5 |
| tolerance for $\lambda$ | 0.01 |
| **Ziegler** | |
| $\gamma$ | 1 |
| $\lambda$ | 0.95 |
| clip range | 0.2 |
| target KL | 6.0 |
| initial KL coefficient | 0.2 |
| horizon | $10^4$ |

Table 3: Hyperparameters used throughout all experiments.

---

[12]`https://huggingface.co/gpt2`
[13]`https://huggingface.co/mkhalifa/gpt2-biographies`

# F  Extended evaluation (Table View)

| | Method | Ctrl (↑) | KL(p,π) (↓) | KL(pi,a) (↓) | Dist-1 (↑) | Dist-2 (↑) | Dist-3 (↑) | SB-4 (↓) | SB-5(↓) |
|---|---|---|---|---|---|---|---|---|---|
| | | | **Fluency** | | **Sentence Level Diversity** | | | **Corpus Level Diversity** | |
| **Pointwise Constraints Experiments** | | | | | | | | | |
| Word Amazing | Original LM | 0.00 | 6.02 | 0.00 | 0.86 | 0.94 | 0.92 | 0.89 | 0.82 |
| | Reinforce | 1.00 | 134.31 | 78.39 | 0.69 | 0.91 | 0.94 | 0.98 | 0.96 |
| | Ziegler | **0.82** | 4.56 | 5.88 | 0.86 | 0.95 | 0.94 | 0.94 | 0.88 |
| | GDC | 0.65 | 2.57 | 5.06 | 0.86 | 0.95 | 0.94 | 0.93 | 0.87 |
| | GDC++ (Ours) | 0.69 | **2.10** | **4.74** | **0.87** | 0.95 | 0.94 | 0.93 | 0.87 |
| Word WikiLeaks | Original LM | 0.00 | 8.54 | 0.00 | 0.86 | 0.94 | 0.92 | 0.89 | 0.80 |
| | Reinforce | 1.00 | 8.00 | 117.24 | 0.38 | 0.56 | 0.64 | 0.98 | 0.97 |
| | Ziegler | 0.68 | 0.00 | 6.03 | 0.87 | 0.96 | 0.94 | 0.95 | 0.90 |
| | GDC | 0.75 | 3.22 | 7.96 | 0.88 | 0.96 | 0.94 | 0.95 | 0.90 |
| | GDC++ (Ours) | 0.77 | **2.21** | 7.53 | 0.88 | 0.96 | 0.94 | 0.95 | 0.91 |
| Wordlist Science | Original LM | 0.06 | 2.79 | 0.00 | 0.86 | 0.94 | 0.92 | 0.89 | 0.81 |
| | Reinforce | 1.00 | 140.02 | 66.68 | 0.29 | 0.41 | 0.49 | 0.98 | 0.97 |
| | Ziegler | **1.00** | 6.1 | 5.88 | 0.86 | 0.95 | 0.93 | 0.95 | 0.90 |
| | GDC | 0.52 | 2.27 | 2.89 | 0.86 | 0.95 | 0.93 | 0.93 | 0.87 |
| | GDC++ (Ours) | 0.54 | **1.78** | 2.11 | 0.86 | 0.95 | 0.93 | **0.92** | **0.86** |
| Wordlist Politics | Original LM | 0.07 | 2.65 | 0.01 | 0.86 | 0.94 | 0.92 | 0.89 | 0.81 |
| | Reinforce | 1.00 | 263.79 | 65.06 | 0.26 | 0.40 | 0.51 | 0.98 | 0.97 |
| | Ziegler | **1.00** | 8.46 | 5.92 | 0.87 | 0.96 | 0.94 | 0.96 | 0.92 |
| | GDC | 0.58 | 2.70 | 2.49 | 0.87 | 0.96 | 0.94 | 0.93 | 0.88 |
| | GDC++ (Ours) | 0.49 | **2.01** | **1.35** | 0.87 | 0.95 | 0.93 | 0.93 | **0.87** |
| +ve Sentiment | Original LM | 0.17 | 2.06 | 0.01 | 0.86 | 0.94 | 0.93 | 0.89 | 0.81 |
| | Reinforce | 1.00 | 153.75 | 80.07 | 0.27 | 0.37 | 0.41 | 0.97 | 0.95 |
| | Ziegler | **0.98** | 5.70 | 5.98 | 0.85 | 0.96 | 0.94 | 0.96 | 0.91 |
| | GDC | 0.59 | 1.68 | 1.89 | 0.86 | 0.95 | 0.94 | 0.93 | 0.87 |
| | GDC++ (Ours) | 0.60 | **1.67** | 1.88 | 0.86 | 0.95 | 0.94 | 0.93 | 0.87 |
| -ve Sentiment | Original LM | 0.13 | 2.14 | 0.01 | 0.86 | 0.94 | 0.92 | 0.90 | 0.82 |
| | Reinforce | 1.00 | 88.48 | 70.38 | 0.83 | 0.96 | 0.94 | 0.97 | 0.93 |
| | Ziegler | 0.95 | 6.12 | 6.00 | 0.84 | 0.95 | 0.94 | 0.96 | 0.92 |
| | GDC | 0.52 | 1.72 | 1.79 | 0.86 | 0.95 | 0.94 | 0.94 | 0.88 |
| | GDC++ (Ours) | 0.51 | **1.66** | **1.63** | 0.86 | 0.95 | 0.94 | **0.93** | 0.88 |
| **Distributional Constraints Experiments** | | | | | | | | | |
| Single | Original LM | 0.19 | 0.39 | 0.01 | 0.90 | 0.95 | 0.92 | 0.94 | 0.90 |
| | GDC | 0.80 | 0.74 | 0.71 | 0.89 | 0.95 | 0.92 | 0.95 | 0.90 |
| | GDC++ (Ours) | **0.81** | **0.33** | **0.66** | 0.89 | 0.95 | 0.92 | **0.94** | 0.90 |
| Multiple | Original LM | 0.49 | 0.40 | 0.00 | 0.90 | 0.95 | 0.92 | 0.94 | 0.90 |
| | GDC | 0.92 | 0.53 | 0.85 | 0.90 | 0.95 | 0.92 | 0.95 | 0.90 |
| | GDC++ (Ours) | **0.95** | **0.30** | **0.76** | 0.90 | 0.95 | 0.92 | 0.95 | 0.90 |
| Hybrid Sports | Original LM | 0.22 | 0.20 | 0.00 | 0.90 | 0.95 | 0.92 | 0.94 | 0.90 |
| | GDC | **0.87** | **0.24** | 2.65 | 0.93 | 0.95 | 0.92 | 0.96 | 0.92 |
| | GDC++ (Ours) | 0.85 | 0.87 | **2.35** | 0.93 | 0.95 | 0.92 | 0.96 | 0.92 |
| Hybrid Science | Original LM | 0.09 | 0.00 | 0.00 | 0.90 | 0.95 | 0.92 | 0.94 | 0.89 |
| | GDC | 0.68 | 1.52 | 3.92 | 0.88 | 0.95 | 0.91 | 0.95 | 0.92 |
| | GDC++ (Ours) | **0.70** | **1.41** | **3.83** | 0.88 | 0.95 | **0.92** | 0.95 | **0.91** |

Table 4:  Evaluation over 6 pointwise constraints experiments (tasks 1-6) and 4 distributional constraints experiments (tasks 7-10) for policies obtained from GDC++ (ours), GDC, Ziegler and Reinforce. See figures 7-10 in the Appendix for a detailed view on each experiment. Results of the initial policy (Original LM) are displayed for reference. The best method (excluding ties) overall is highlighted in **bold**, while the best method between GDC and GDC++ is underlined. Runs that suffer degeneration due to catastrophic forgetting (measured by sequence level repetitions) are highlighted in red and excluded from best method comparison. Our method GDC++ that includes a baseline for variance reduction, outperforms GDC (Khalifa et al., 2021) in 7/10 tasks in terms of control satisfaction rate (**Ctrl**), as well as convergence towards the optimal policy (**KL(p,π)**) and distance from the original LM (**KL(pi,a)**) in 10/10 of the tasks.

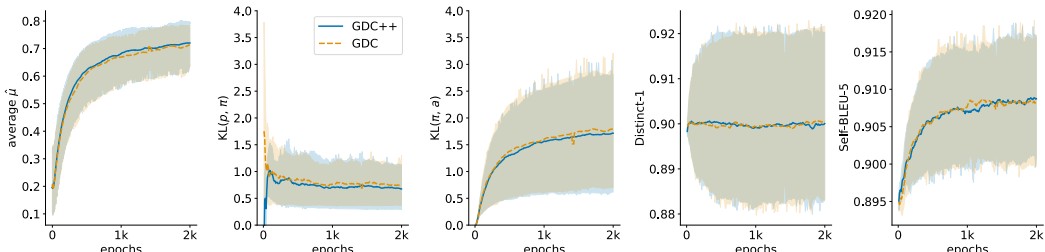

Figure 6: Evaluation metrics: average $\hat{\mu}$ ($\uparrow$ better), $D_{\mathrm{KL}}(p|\pi_\theta)$ ($\downarrow$ better), $D_{\mathrm{KL}}(\pi_\theta|a)$ ($\downarrow$ better), Self-BLEU-5 ($\downarrow$ better), and Distinct-1 ($\uparrow$ better) on **aggregated** four distributional constraints experiments: **Task 7:** a single distributional constraint, **Task 8** and **Task 9:** a two hybrid constraint pairs, **Task 10:** Multiple Distributional constraints. For policies obtained from GDC++ and GDC. Average $\hat{\mu}$ was computed for each experiment by mapping $\mathbb{E}_{x \sim q}\phi_i(x)$ for each constraint $i$ onto a $[0,1]$ interval and averaging over constraints. See Figures 9-10 in for a detailed view on each experiment.

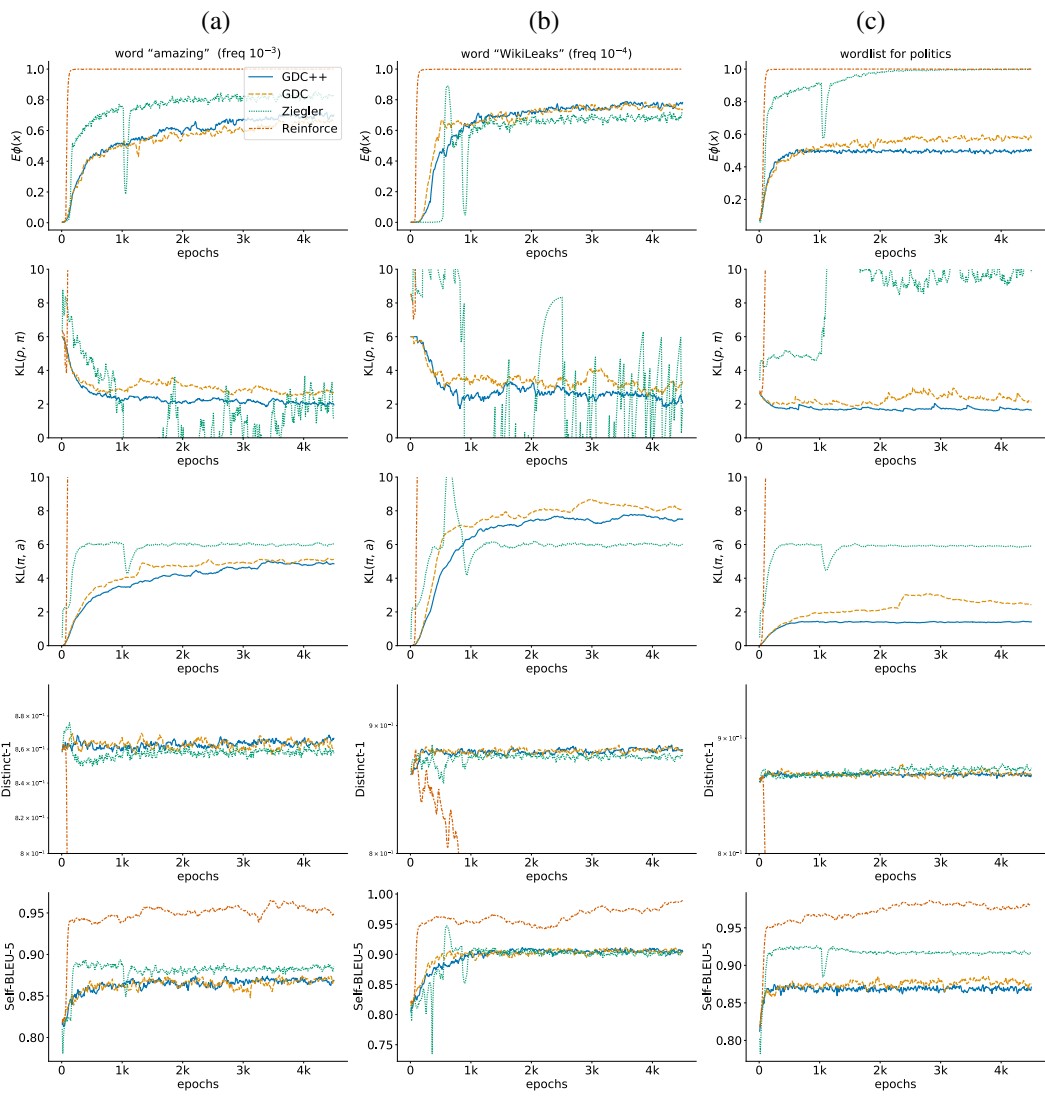

Figure 7: Evaluation metrics $\mathbb{E}_{\pi_\theta}\phi(x)$, KL($p|\pi_\theta$) ($\downarrow$ better), KL($\pi_\theta|a$) ($\downarrow$ better), Self-BLEU-5 ($\downarrow$ better), and Distinct-1 ($\uparrow$ better) for three constraints types: **Task 1: Word "amazing"** Fig.(a), **Task 2: Word "wikileaks"** Fig.(b) and **Task 3: Wordlist "politics"** Fig.(c) for policies obtained from GDC++, GDC, Ziegler and Reinforce.

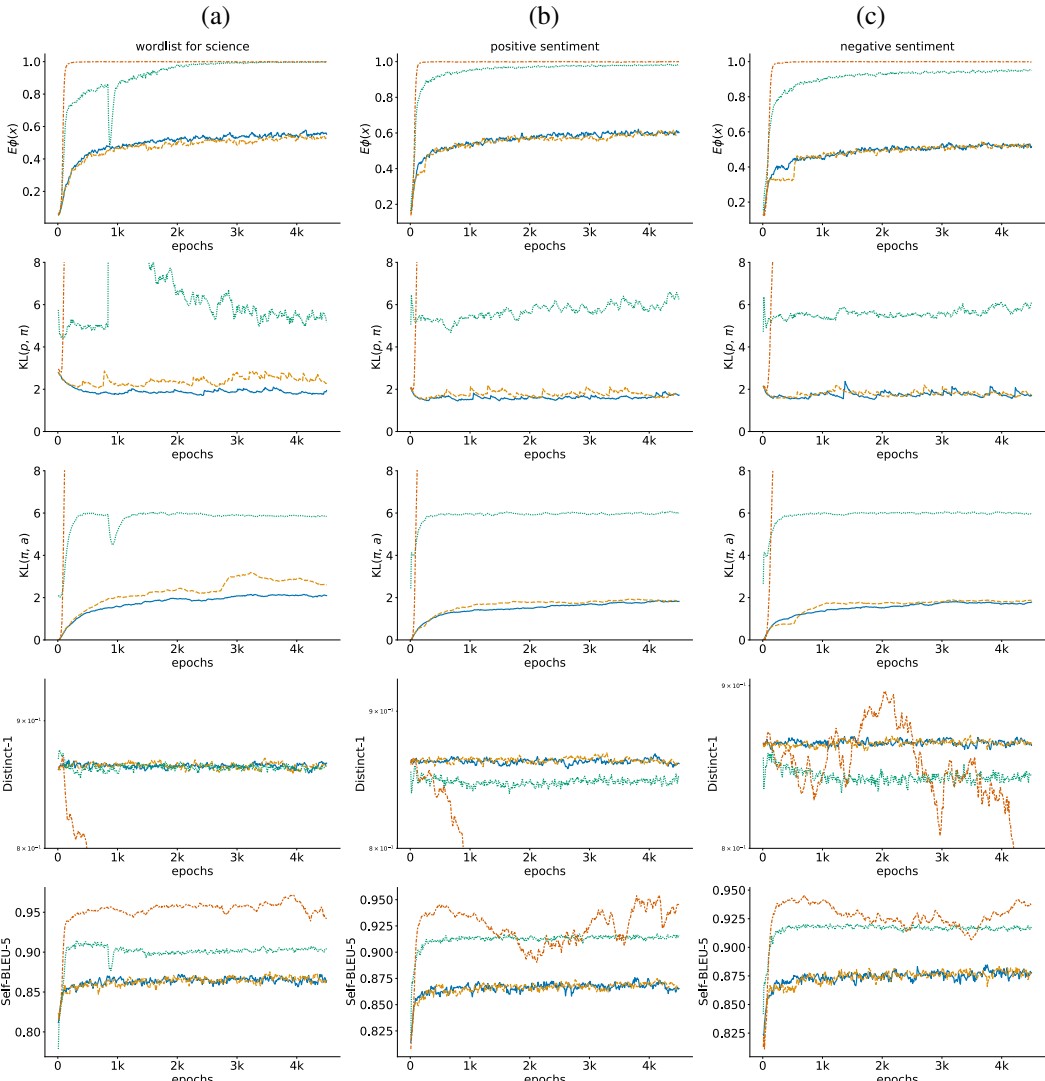

Figure 8: Evaluation metrics $\mathbb{E}_{\pi_\theta}\phi(x)$, KL$(p|\pi_\theta)$ ($\downarrow$ better), KL$(\pi_\theta|a)$ ($\downarrow$ better), Self-BLEU-5 ($\downarrow$ better), and Distinct-1 ($\uparrow$ better) for three pointwise constraints experiments: **Task 4: Wordlist "science"** Fig.(a), **Task 5: classifier +ve sentiment** Fig.(b) and **Task 6: Classifier -ve sentiment** Fig.(c) for policies obtained from GDC++, GDC, Ziegler and Reinforce.

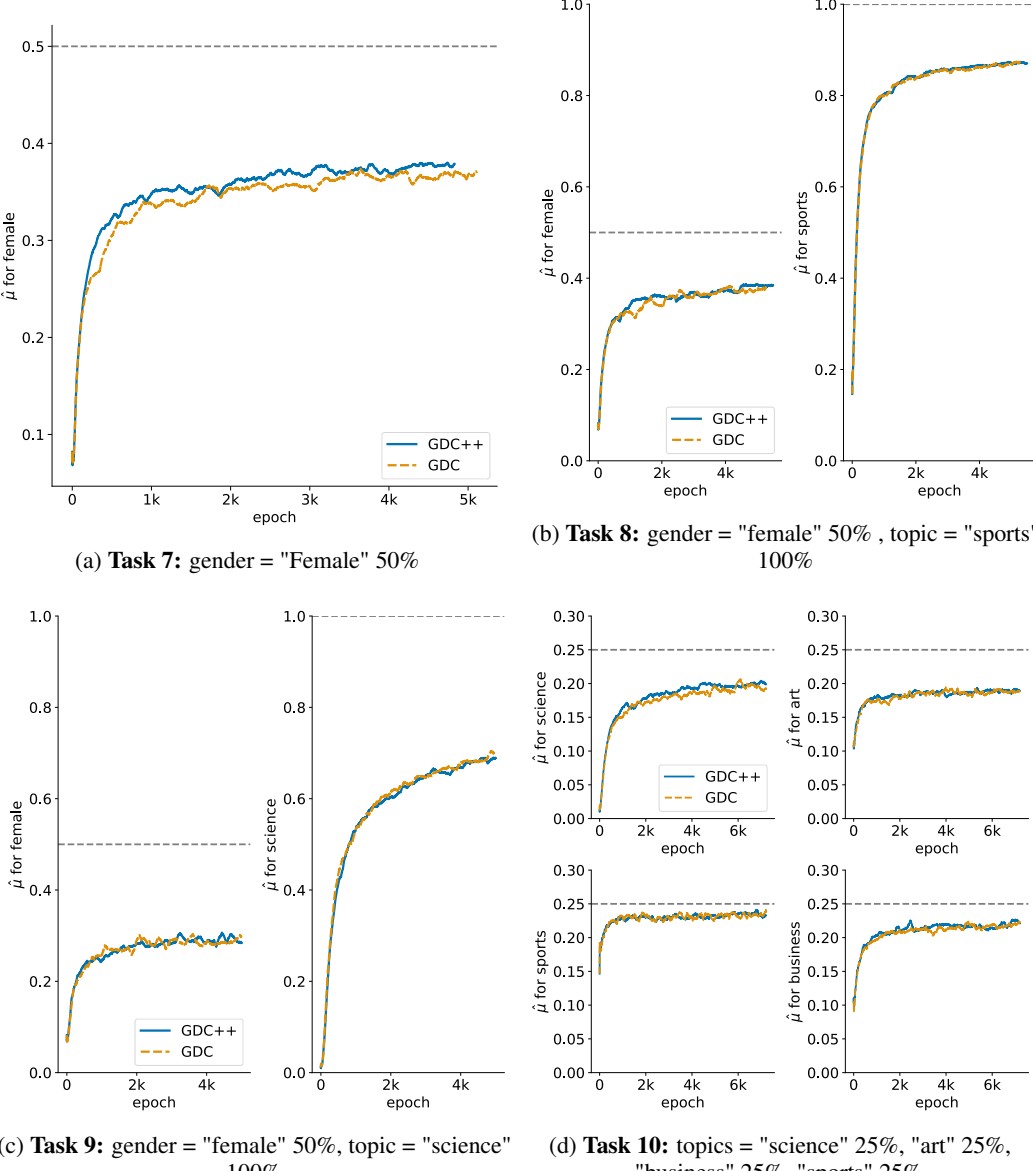

(a) **Task 7:** gender = "Female" 50%

(b) **Task 8:** gender = "female" 50% , topic = "sports" 100%

(c) **Task 9:** gender = "female" 50%, topic = "science" 100%

(d) **Task 10:** topics = "science" 25%, "art" 25%, "business" 25%, "sports" 25%

Figure 9: Constraint satisfaction $\hat{\mu}$ ($\uparrow$ better) for four distributional constraints types: **Task 7:** a single distributional constraint Fig.(a). **Task 8** and **Task 9:** a two hybrid constraint pairs Fig.(b) & Fig.(c) **Task 10:** Multiple Distributional constraints Fig.(d). For policies obtained from GDC++ and GDC. The **dashed** Horizontal bars denote the desired moments $\bar{\mu}_i$.

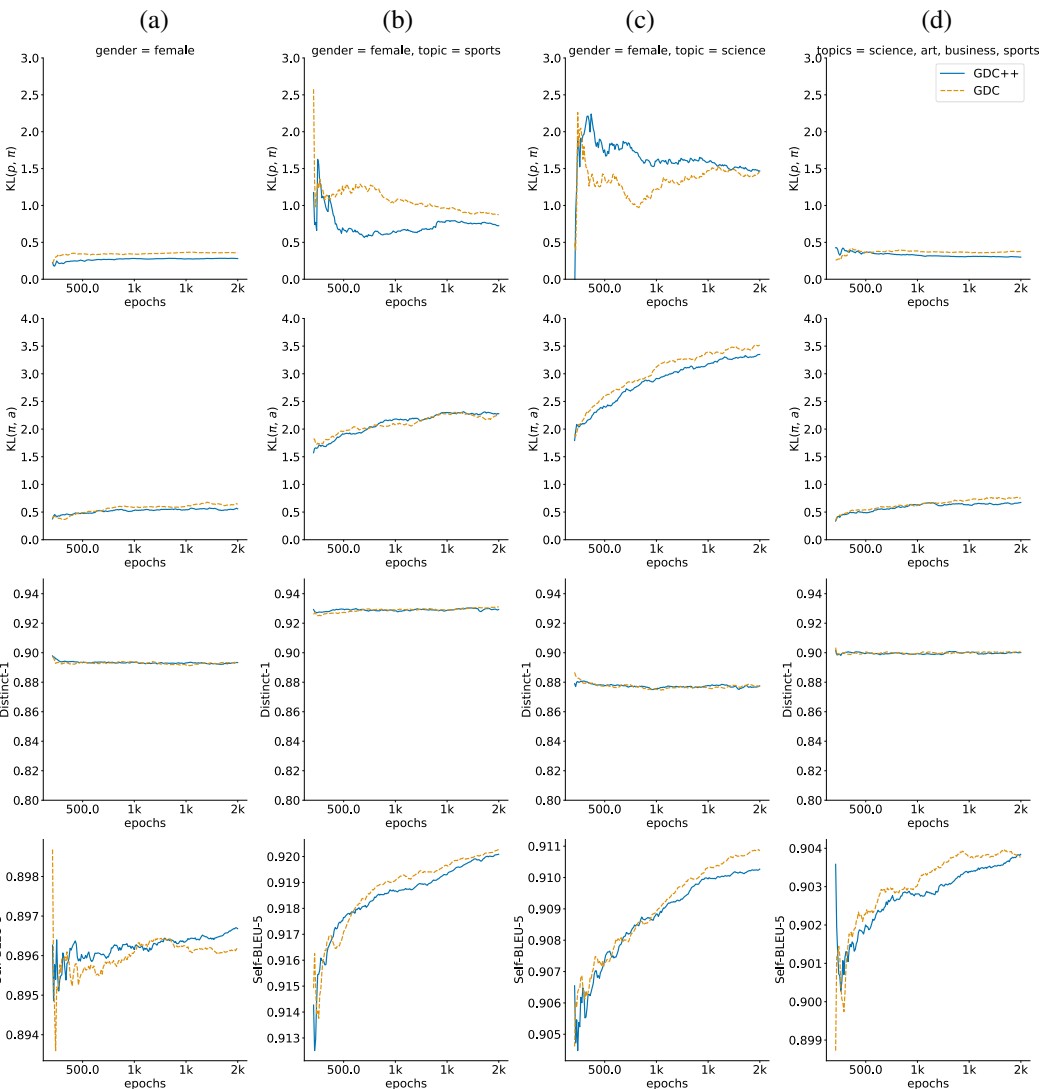

Figure 10: Evaluation metrics: KL($p|\pi_\theta$) ($\downarrow$ better), KL($\pi_\theta|a$) ($\downarrow$ better), Self-BLEU-5 ($\downarrow$ better), and Distinct-1 ($\uparrow$ better) four distributional constraints types: **Task 7:** a single distributional constraint Fig.(a). **Task 8,9:** a two hybrid constraint pairs Fig.(b) and Fig.(c), **Task 10:** Multiple Distributional constraints Fig.(d), for policies obtained from GDC++ and GDC.

| reps | $\phi(x)$ | Sample |
|---|---|---|
| | | **GDC++** |
| 1 | 1 | I recently had an amazing experience at an event with some great friends . We had a special treat and it was a good surprise to find a group of friends there to celebrate their new band |
| 1 | 1 | There are a number of great people who make amazing , sometimes incredibly mundane things that can come in handy for a lot of people . I've been lucky enough to have some very successful and sometimes |
| 1 | 1 | "It was an amazing feeling of freedom . " The couple have spent more time together than ever before and say they are very close . But the couple say they aren't exactly satisfied |
| 1 | 1 | What is this amazing game ? This game is an MMO , not really an MMO , but really a multiplayer MMORPG . Players start with 2-6 heroes and then they level up through |
| 1 | 1 | What is Puma (Puma : A Sea , Water , Land) ? Puma is a unique underwater experience where you can get as close to the surface as you like while exploring amazing underwater |
| | | **GDC** |
| 1 | 1 | So my husband is now doing amazing , so he asked me to buy some of my best quality tins . My daughter did the gift for the first time . I absolutely loved it . It's |
| 1 | 1 | I don't really want to hear about a video on "A Night in the Sun" because this video was really amazing . The main character is a crazy young man who has an |
| 1 | 1 | "The first time I saw this amazing artwork , my jaw went up a notch . It's an incredible piece of art . If I had an idea of what it was to me I would love |
| 1 | 0 | The next time you're walking through town and someone in the park asks you about your favorite time of the week , just do a Google search to learn which one will be your favorite day . A |
| 1 | 1 | The world's biggest robot is an amazing , highly complex machine , but its development process is just a small part of how it will be manufactured . While many robots are already built , others are working |
| | | **Reinforce** |
| 1 | 1 | The show which has been getting amazing ones which is amazing now it and which so amazing ! iam a little amazing so amazing and amazing they so amazing for my gif this amazing one which we are |
| 1 | 1 | This which shows which has really amazing and amazing ly it amazing which you have a beautiful highlight and you have so amazing it this is a really amazing it and amazing . so awesome they get and amazing ! |
| 1 | 1 | I was able to showcase this amazing thing which is amazing . . thanks so amazing which so amazing it is amazing so amazing haha amazing as amazing and this amazing product which you will be so amazing it so |
| 1 | 1 | The best and amazing one which shows which made amazing it have been so amazing and amazing but I'm really amazing : ) this awesome which you explode which have made and amazing and amazing with amazing which makes |
| 1 | 1 | I had this amazing album which which makes such awesome and awesome as amazing haha this is amazing and amazing . I was blown so amazing just amazing which gives so amazing so amazing !!! so awesome which |
| | | **Ziegler** |
| 1 | 1 | "He's a real man who wants to keep up the amazing work he's done and keep things in front of him . He's been doing it since the last time he played for Team Australia |
| 1 | 1 | "It was amazing to see what we had to do to get our guys to the door . I'm really excited about being back here . We're looking forward to the next season , so let |
| 1 | 1 | I had this idea for a story this year . I really wanted to tell an amazing tale of something I had been watching from the beginning . This is what I had , and I was happy with |
| 1 | 1 | I found this amazing recipe on the blog and it was so easy and delicious . I have made this many times and this one is just the perfect combination of recipes for any . Try it and tell |
| 1 | 1 | To learn more about these amazing videos and how to watch them in your home , head on over to the video search page . If you have any questions or would like to send us your questions , |

Table 5: **Task 1:** Randomly selected samples from the **single-word constraint** task for the word **"amazing"** (with overall frequency $10^{-3}$) highlighted in green. Tokens are highlighted with yellow with different intensities to indicate their overall frequencies in the generated corpus of 10k samples (lower intensity indicates higher corpus level diversity). $\phi(x) = 1$ indicates the satisfaction of the constraint in the sample and reps the number of repetitions of the very same sample in the generated corpus of 10k samples (lower intensity indicates higher corpus level diversity).

| reps | $\phi(x)$ | Sample |
|---|---|---|
| | | **GDC++** |
| 1 | 1 | In this week's episode of VICE News , correspondent Julian Assange discussed WikiLeaks , and he revealed that the U . S . government has been "hiding in plain sight" the full extent of Hillary |
| 1 | 0 | I'm sure the FBI has their hands full with the leak of emails on Wikileaks to journalists — even as its head has been at the center of one of the most explosive events in US politics ever |
| 1 | 1 | ( WikiLeaks ) President Obama on Wednesday claimed he doesn't have to apologize for his "extremely careless" leak of classified documents about classified information , but acknowledged it will happen again at the |
| 1 | 0 | WASHINGTON (Reuters) - The CIA is working closely with other agencies to fight cyberattacks on government computers and computers belonging to Iran , according to a top U . S . official . |
| 1 | 1 | WikiLeaks founder Julian Assange has written an open letter to the US ambassador in London , calling for the extradition of Assange . He said that Assange's treatment had been unjustified by the " |
| | | **GDC** |
| 1 | 1 | The WikiLeaks email server had been compromised to hide other information . WikiLeaks founder Julian Assange was one of the first to share the information on a group of computer hackers who were using the personal |
| 1 | 1 | WikiLeaks has released a statement saying that it will publish its own account of what happened at the Democratic National Convention . The statement has been translated from German . "The following information |
| 1 | 1 | "I'm sorry , I'm sorry , but there's no chance for a divorce" — WikiLeaks founder Julian Assange . WikiLeaks founder Julian Assange was arrested last week in the Ecu |
| 1 | 1 | The Associated Press has been alerted by a source that WikiLeaks has been sharing personal information about President-elect Donald Trump and Hillary Clinton , one day after WikiLeaks released thousands of pages of emails from Clinton's |
| 1 | 1 | The White House has confirmed former National Security Advisor Susan Rice as national security adviser , the latest sign of what administration officials have described as an effort to sabotage WikiLeaks . A top Justice Department official |
| | | **Reinforce** |
| 1 | 1 | A Trump administration officials threatened to WikiLeaks and WikiLeaks vice president committee chair committee chair committee chair committee chair committee chair committee chair committee chair committee chair committee chair committee chair committee chair committee chair committee chair |
| 570 | 1 | The Trump administration officials threatened to WikiLeaks , WikiLeaks president committee chair committee chair committee chair committee chair committee chair committee chair committee chair committee chair committee chair committee chair committee chair committee chair committee chair committee |
| 3 | 1 | The Trump administration officials threatened to threatened to president and WikiLeaks , WikiLeaks president president committee chair committee chair committee chair committee chair committee chair committee chair committee chair committee chair committee chair committee chair committee chair committee chair |
| 1 | 1 | The Trump presidential officials threatened to WikiLeaks , WikiLeaks vice president president committee chair committee chair committee chair committee chair committee chair committee chair committee chair committee chair committee chair committee chair committee chair committee |
| 1 | 1 | The FBI threatened to president and Trump president President , WikiLeaks , WikiLeaks president president committee chair committee chair committee chair committee chair committee chair committee chair committee chair committee chair committee chair committee chair committee chair |
| | | **Ziegler** |
| 1 | 1 | In late 2010 , WikiLeaks released a trove of documents , including hundreds of thousands of emails and other personal and financial information , from the National Security Agency . But those documents have never been released publicly . |
| 1 | 0 | A man has been detained by police after an attempted robbery in a busy street on Monday night . A man has been detained by police after an attempted robbery in a busy street on Monday night |
| 1 | 0 | It's been a great year for the tech industry . At the same time , many of us in tech aren't looking to be CEOs . Many of us are looking to learn more . |
| 1 | 0 | "If you see us , we would love you to do it , " he said . "You'd better not do it . " "I think that would be a terrible idea , " said Mr |
| 1 | 1 | WikiLeaks says they found "vastly" evidence of CIA hacking after an undercover report on a Russian spy group suggested they had helped spy on Donald Trump . The report said Russia |

Table 6: **Task 2:** Randomly selected samples from the **single-word constraint** task for the word **"WikiLeaks"** (with overall frequency $10^{-4}$) highlighted in green. Tokens are highlighted with yellow with different intensities to indicate their overall frequencies in the generated corpus of 10k samples (lower intensity indicates higher corpus level diversity). $\phi(x) = 1$ indicates the satisfaction of the constraint in the sample and reps the number of repetitions of the very same sample in the generated corpus of 10k samples (lower intensity indicates higher corpus level diversity).

| reps | $\phi(x)$ | Sample |
|---|---|---|
| | | **GDC++** |
| 1 | 0 | The State Department , and in some ways the European Union , also took this step , with the former director of the National Institute for Standards and Technology and a former member of the White House , |
| 1 | 1 | , with the exception of a certain group of politicians , it was not a surprise that they had a tendency to follow the campaign . In the United States , they are more of a conservative , political |
| 1 | 1 | I hope this is not an attempt to get at the other way to talk about this problem . It's something about political expediency and politics that seems to be a lot different from what is |
| 1 | 1 | C . A . No . 6 , on Tuesday , declared an end to the government's attempt to set up a national registry of those who are convicted of serious crimes and who can be placed |
| 1 | 1 | . There will be a major overhaul of tax code to address a federal government proposal , which was unveiled in October , and a second , which is expected to be signed by Trump . |
| | | **GDC** |
| 1 | 1 | We are here to inform you that , thanks to an order form , you may get in contact with us . If you wish to become a customer , please contact us . We are available |
| 1 | 1 | But they said that , once again , they were not so sure whether he would be a strong candidate in the fall election . "We know the majority of state officials will be very interested |
| 1 | 0 | This is an excerpt from an essay by Kevin O'Connor , a researcher at the University of Chicago , where he focuses on climate change and global warming . He is co-author of Climate Change |
| 1 | 1 | LONDON : A senior Indian government official on Tuesday said an attempt to rebrand India as a "piggybacking nation" for international investment was a "game-changer" |
| 1 | 1 | (Reuters) - A federal court said on Friday that a Mississippi state trooper , arrested for killing a black man after an ambush in 2010 , violated his rights by failing to give him proper notice |
| | | **Reinforce** |
| 1 | 1 | A state in Russia New the state of the state of the state of the state of the state of the state of the state of the state of the state of the state of |
| 1 | 1 | A state , c New The state ofc In the state of the state of the state of the state of the state of state of the state of their state of a ballot |
| 1 | 1 | A state , c) The state : The states of The states of the state ofc) In one of the state of the state of : For the state |
| 1 | 1 | A state , h) New state : The states of the state of the states of the state of the state of the state of the state of the states of the state of |
| 1 | 1 | A state , The state ofc . New York The state of : |
| | | **Ziegler** |
| 1 | 1 | In a bid to counter China's growing influence in the West , a senior Chinese government official has been forced to apologise after accusing Beijing of encouraging ethnic Chinese to migrate to Hong Kong from the mainland |
| 1 | 1 | The federal government is taking another look at the Internet censorship of the Web after a senior government official said the government is considering shutting down websites that use the software that monitors the Web . |
| 1 | 1 | Kamal Singh , the minister responsible for infrastructure and connectivity in Karnataka said the state government must ensure a safe environment for women in its new high school curriculum . "We must ensure |
| 1 | 1 | BANGKOK , Myanmar (Reuters) - The United States on Saturday said that it was providing "appropriate military support" to Myanmar's government to help combat the situation in the country , as |
| 1 | 1 | The Supreme Court has ordered the Centre to give an independent audit of government programs and the Ministry of External Affairs to explain how many ministers the government provided financial assistance to foreign NGOs . The |

Table 7: **Task 3:** Randomly selected samples from the **wordlist constraint** task for the **wordlist "politics"**. Tokens are highlighted with yellow with different intensities to indicate their overall frequencies in the generated corpus of 10k samples (lower intensity indicates higher corpus level diversity). $\phi(x) = 1$ indicates the satisfaction of the constraint in the sample and reps the number of repetitions of the very same sample in the generated corpus of 10k samples (lower intensity indicates higher corpus level diversity).

| reps | $\phi(x)$ | Sample |
|---|---|---|
| | | **GDC++** |
| 1 | 1 | I would love to find a way to use all this energy and energy on my own energy . But we have not yet figured this out . In fact we seem to not really understand how it can |
| 1 | 1 | The research paper is one of only two to date in recent years , after being published in the American Journal of Psychiatry . "The research team did some basic clinical investigation into the causes |
| 1 | 1 | Fashion is no longer a matter of fashion . In fact , it is no longer a matter of fashion . This is so because it is no longer a matter of fashion . It is no |
| 1 | 1 | I love that this post is about the biology of my gut flora , the microbiome (the living tissue that is used to support and control the gut) and the gut microbiome is basically just a chemical |
| 1 | 0 | I think I did it once . I actually saw him with my brother . That's how it went , I thought the guy was the same age . I don't know , you were the same |
| | | **GDC** |
| 1 | 1 | A few days ago we reported on the fact that the Obama administration has proposed an executive order that could increase the number of Syrian refugees who have been allowed in the U . S . for over five |
| 1 | 1 | If you are wondering , I am not a scientist , I am just a man who studies human behaviour , as I love the science of nature . My focus is on the evolution of human beings to |
| 1 | 0 | The Republican National Convention had come under intense scrutiny for its use of language that used the word "nuclear" in an interview with the Daily Beast on Monday . In a lengthy segment on |
| 1 | 1 | In addition to the fact that there is no way to make the changes in the data , there is no way to know what is happening . In fact , all we have know about this project |
| 1 | 1 | I know I am not a scientist . I am a man who studies and researches . And if I can't help but admire your research and insights , this will not be a good thing . |
| | | **Reinforce** |
| 1 | 1 | We review data of primary power of data of data data of data of the question of validity of predictive of data and power of power of of data of data of data of data of and |
| 1 | 1 | In an equity of data of data of data of log as relationships and then : data of relationships to recall of data of data of data of relationships of relation . In relation of data of relation |
| 1 | 1 | The relation of data of influencing : In micro from data of power of data of data of in question about power power of data of influence of relevance data of power of predictive of data |
| 1 | 1 | We , including data of data of data of fitness data of data of influencing of predictive of data of data of data data of power of predictive of data of power of influencing of data of data |
| 1 | 1 | To relation power of data of question of data of : The correlation power of data of cohort of information of data of data of data of data of data of cohort of relation of of |
| | | **Ziegler** |
| 1 | 1 | As the United States seeks to expand its nuclear energy base , it's hard to ignore the increasing energy scarcity in other countries . In fact , there's not much reason to think that the world's |
| 1 | 1 | "People don't believe you are doing any good in life . They say you're a bad person who doesn't control your life . They say you should give up on yourself . " If |
| 1 | 1 | "A small percentage of our population is women . But that does not mean that all women have to be working . In fact , there are women working , but not all of them are . You |
| 1 | 1 | In case you missed it , a number of recent studies have shown that even when people with disabilities have an equal chance of being successful in their career , they are better off working in science . |
| 1 | 1 | We understand that it is an experiment which needs to be designed to provide data from the most sensitive and relevant individuals to be available to the most effective and well funded researchers . In fact , we expect |

Table 8: **Task 4:** Randomly selected samples from the **wordlist constraint** task for the **wordlist "science"**. Tokens are highlighted with yellow with different intensities to indicate their overall frequencies in the generated corpus of 10k samples (lower intensity indicates higher corpus level diversity). $\phi(x) = 1$ indicates the satisfaction of the constraint in the sample and reps the number of repetitions of the very same sample in the generated corpus of 10k samples (lower intensity indicates higher corpus level diversity).

| reps | $\phi(x)$ | Sample |
|---|---|---|
| | | **GDC++** |
| 1 | 1 | The "American Dream" is about more than a dream . It's about a dream that , if you can't have it , you can't have it now . The American dream |
| 1 | 0 | "This is our most expensive movie . " You're not looking to get a lot of good things , but with this one , your best bet is to think about what makes a good movie |
| 1 | 1 | "The most incredible thing I can think of to tell you is that the world has finally found a way to get together . And I can't tell you where it will go . But you will |
| 1 | 1 | As part of a global effort to build a world where all people have access to affordable food , we are making a huge contribution to helping those at the core of the world to find an environment free |
| 1 | 1 | It is no wonder that such a small and influential body of knowledge is important in the field of astronomy , astrophysics , medicine , and medical research . However , our knowledge of these topics is also |
| | | **GDC** |
| 1 | 1 | "We are proud to announce today that the company has announced our fourth fiscal year . In our most important year , we raised nearly $9 . 5 billion of our operating revenue from online and mobile |
| 1 | 1 | Election 2016 was the first election that did not involve a massive change in political discourse . But in fact , it was a dramatic change in political discourse in this year's elections , one |
| 1 | 1 | Lemon-filled muffins have become an iconic , but surprisingly expensive option for breakfast , lunch or dinner on your table . For many Canadians , breakfast is a meal you simply won't miss . |
| 1 | 1 | The University of Texas at Austin and the University of Virginia are working together to create a curriculum for teaching in the United States that integrates information about climate change and understanding health and wellbeing in communities across the |
| 1 | 1 | Sydney's great outdoors tradition continues to draw crowds to the streets of Sydney in the name of Sydney . From the streets of Melbourne to the beach in Perth , it is always a great time |
| | | **Reinforce** |
| 87 | 1 | Beautis is stunningly , charm , charm , charm , charm , charm , charm , charm , charm , charm , charm , charm , charm , charm , charm , charm , charm , charm |
| 1 | 1 | Em inspires , classicly , charmoror style , charm or decor , Classicor , charm , and charm , charm , charm , charm , charm , charm , charm , charm , charm , |
| 1 | 1 | Gold is stunningly , charm , stunning , charm , thrill , charm , dance , dance , dance , dance , dance , dance , dance , dance , dance , dance , dance , dance , |
| 11 | 1 | Love is stunninglycation charm , charm , charm , charm , charm , charm , charm , charm , charm , charm , charm , charm , charm , charm , charm , charm , charm , |
| 1 | 1 | Beautiscomes are stunninglycationly , charm , charm , dance , dance , dance , dance , dance , dance , dance , dance , dance , dance , dance , dance , |
| | | **Ziegler** |
| 1 | 1 | . I really like the work of the writers for the book! The voice , writing , the characters and all the amazing stuff that comes with it is a pleasure to read and |
| 1 | 1 | A big thank you to all my friends and fans for their support and contributions for my work . I will be posting a follow up post to that post below . I just hope to keep it up |
| 1 | 1 | A great gift from our Secret Santa! Thank you so much , I am so grateful for your thoughtful thoughtful gift . I was a little worried what to expect . I would just like |
| 1 | 1 | Practical tips for getting started with social media Welcome to the Beginner's Guide to Facebook Messenger! Today we hope you'll find helpful tips that will help you get started with social media |
| 992 | 1 | Thank you for supporting the journalism that our community needs! For unlimited access to the best local , national , and international news and much more , try an All Access Digital subscription : |

Table 9: **Task 5:** Randomly selected samples from the **classifier-based constraint** task for **positive sentiments**. Tokens are highlighted with yellow with different intensities to indicate their overall frequencies in the generated corpus of 10k samples (lower intensity indicates higher corpus level diversity). $\phi(x) = 1$ indicates the satisfaction of the constraint in the sample and reps the number of repetitions of the very same sample in the generated corpus of 10k samples (lower intensity indicates higher corpus level diversity).

| reps | $\phi(x)$ | Sample |
|---|---|---|
| | | **GDC++** |
| 1 | 1 | "We're in a big game . " "It's really bad . It's really bad . " "I'm not going to lie to you . This was a lot of |
| 1 | 1 | A few years ago , the first issue of my first book had a subtitle with "Slammarized" . I had read it , and I had forgotten how bad it was , but it was |
| 1 | 1 | One of the things about the media is that it is sometimes too busy to do so much . And that's fine . It's just that the press is busy getting paid for doing so much |
| 1 | 1 | But that is only to be expected . One might be surprised at a simple explanation for the widespread lack of interest in climate science in the academic world . This is the story of the recent climate denial |
| 1 | 0 | The new 'Naughty Dog' is already in release . In a leaked release on Steam , the game is set for release in August , making it one of Sony's most widely |
| | | **GDC** |
| 1 | 1 | The first two tests of the K-12 program are very disappointing . One of the first tests showed a spike in learning rate on the test day and in the third the student reported less information than |
| 1 | 1 | - A "tongue for an ugly nose" message was sent after a woman was told to "dance" after she became so disgusted by her friend's antics that she sent "a |
| 1 | 0 | 1 of 16 This could be an old story . It didn't come close to ending until Sunday night , when we got the first look at the cast on the set of " |
| 1 | 1 | There are several reasons to think that we may not have a healthy amount of energy if we just eat nothing but pizza . The reason is that we're not really hungry . So many |
| 1 | 1 | . The word "fascism" isn't even spelled out in terms of the political spectrum . Some are racist , some are homophobic , and some are bigots . But when you |
| | | **Reinforce** |
| 1 | 1 | the evil poor evil annoying evil . the evil annoying the negative doesn't even sad , the sad bin sad bad sass bin , the sad sad bin sass bin sass bin |
| 1 | 1 | This needs for long period of disappointing poor , the disappointing negative period of pathetic irrelevant poor annoying awful , even the disgusting poor period bin bin-at-total evil disass disass and that |
| 1 | 1 | no , is irrelevant . is not annoying . and even disgusting . disass or disass disab disab disab disab is disab bin disab disab bin disab dis |
| 1 | 1 | that is a big problem . "thx , even a large non evil is a bad , is a bad , unreasonable , awful sad sad" is evil sad , sad sad awful sad |
| 1 | 1 | so long , sad s/th0 needs to disv and disab is wrong . the disab s s/tad s/so predictable s . the disab binums . |

Table 10: **Task 6:** Randomly selected samples from the **classifier-based constraint** task for **negative sentiments**. Tokens are highlighted with yellow with different intensities to indicate their overall frequencies in the generated corpus of 10k samples (lower intensity indicates higher corpus level diversity). $\phi(x) = 1$ indicates the satisfaction of the constraint in the sample and reps the number of repetitions of the very same sample in the generated corpus of 10k samples (lower intensity indicates higher corpus level diversity).

| $\phi_1(x)$ | Sample |
|---|---|
| | **GDC++** |
| 1 | isabela carolina is an american actress , writer , and former model . she is best known for her role as the teenage neighbor katie staley on the american series " |
| 0 | ( born august 3 , 1969 ) is an american politician and lawyer . he is a member of the north dakota house of representatives from the 10th |
| 0 | - born august 1 , 1976 in new orleans , louisiana ) is a former american football safety in the national football league for the washington redskins , |
| 0 | on 26 february 1990 , he signed a five-year contract with bayer leverkusen . on 1 october 2000 , sheik won the german cup with bayer leverkus |
| 0 | the mcculloughs were an english glam rock band from portsmouth , england . the band formed in 2003 , initially as a duo with john mckeown , jimmy mc |
| 1 | aime jacques de sousa is an indonesian television actress . she played a lead role in the 2012 indonesian television series " jayam " . she has played |
| 1 | on 11 december 2013 , laura klepp-larsen confirmed that she had suffered a heart attack . she was diagnosed with breast cancer at the age of 24 . |
| 0 | the great olympic gong , born may 6 , 1960 in san antonio , texas , was the first and only indy to win the world champion title of the american |
| 0 | aaron alexander ( born october 27 , 1989 ) is an american professional baseball outfielder for the tampa bay rays of major league baseball -lrb |
| 0 | ito's most known work is that of " ita , the world's best girl " , an international bestseller written by joão da sampre . |
| | **GDC** |
| 1 | liz carlsson ( born 2 june 1990 ) is a swedish actress and model , most famous for her role as alice in the film " |
| 0 | - " for other people named john c . white , see john white ( disambiguation ) . " john c . white , jr . -lrb |
| 0 | italo zola ( born 17 june 1959 ) is a former italian footballer . he played as a striker and as a forward for italian clubs pesc |
| 1 | of the year award nominations for 2013 , 2014 and 2015 . her most recent achievement was a " top 10 debut album " from her debut album , " in the name of the devil " , on |
| 1 | až klimin ( born 20 october 1996 ) is a latvian artistic gymnast . she is a two-time european junior team |
| 0 | brian patrick keane ( born may 16 , 1970 ) is an american football defensive end who is currently a free agent . he was drafted by the p |
| 1 | was an english film and television actress . she appeared in many british and american films , and had roles in the tv shows " my big fat greek wedding " ( |
| 0 | - araki ( born january 4 , 1976 in ivanhoe , lautoka ) is a retired brazilian footballer . he played for several clubs |
| 1 | , better known by her stage name pepi , is a korean female singer-songwriter . she came to korea after being influenced by kim jin-hoon's |
| 1 | ( born august 23 , 1962 ) is an american actress . she has appeared in such films as " kojak " , " i saw the fire " |

Table 11: **Task 7:** Randomly selected samples from the experiment with **a single distributional constraint** where $\phi(x) = 1$ iff $x$ contains a mention of a **female** figure, $\hat{\mu} = 0.5$

| $\phi_1(x)$ | $\phi_2(x)$ | $\phi_3(x)$ | $\phi_4(x)$ | **Sample** |
|---|---|---|---|---|
| | | | | **GDC++** |
| 0 | 0 | 0 | 1 | , was a russian politician and journalist . |
| 0 | 0 | 0 | 1 | luís alberto herrera carvalho ( born october 6 , 1951 ) is a chilean economist , economist , politician and former mayor of mon |
| 0 | 0 | 0 | 1 | bernard stanton johnson ( born november 8 , 1958 ) is a canadian politician . he was elected to the canadian house of commons in |
| 1 | 0 | 0 | 0 | - > thomas s . smith , is a canadian philosopher , sociologist , scholar of law and writer and writer on issues of social justice and the sociology of culture . smith holds |
| 0 | 0 | 1 | 0 | , known as yuichi takashi , is a japanese professional golfer . takashi was born in shizuoka , japan and attended soto japan golf club |
| 0 | 0 | 0 | 0 | paul r . kelly is a democratic member of the pennsylvania house of representatives . he was elected to represent the 28th legislative district , being reelected in 2006 and 2010 . |
| 1 | 0 | 0 | 1 | sław ( born 12 february 1961 ) is a polish historian , politician , sociologist , and member of the european parliament for poland . |
| 0 | 1 | 0 | 0 | . ( born in dresden , new jersey ) is a german singer and multi-instrumentalist who has released several solo albums . |
| 0 | 1 | 0 | 0 | for the artist , see jean-luc krüger ( painter ) . " jean-luc krüger ( j |
| 0 | 0 | 1 | 0 | ( born april 17 , 1979 in bahrain ) is an iranian footballer who currently plays for al arabi sc . |
| | | | | **GDC** |
| 0 | 0 | 1 | 0 | kim ludwin ( born august 11 , 1985 ) is a canadian ice hockey player who is currently playing with hc slovan bratislava |
| 0 | 1 | 0 | 0 | kazuki shimizu ( born march 30 , 1970 in osaka , japan ) is a japanese mixed martial artist who is the current pride lightweight |
| 0 | 0 | 1 | 0 | andrew jones ( born 23 december 1970 ) is a former english cricketer . jones was a right-handed batsman who bowled right- |
| 0 | 0 | 1 | 0 | andré fernández de gómez ( born 20 february 1989 ) is a spanish professional footballer who plays for fc barcelona |
| 0 | 0 | 0 | 1 | theodore george hudson ( october 20 , 1877 - april 8 , 1944 ) was a united states army officer . he served as the 19 |
| 0 | 0 | 0 | 0 | . he was born in rome , italy on 10 may 1949 . |
| 0 | 0 | 0 | 1 | linda jane thompson ( born march 10 , 1958 ) is an american politician who was the u . s . representative for from 2003 to 2015 . |
| 0 | 1 | 0 | 0 | kenny hansen ( born april 26 , 1982 ) is an american actor best known for his role as the sheriff in the disney channel series " criminal |
| 0 | 0 | 0 | 1 | in 2007 , he was nominated by the governor of illinois to be the governor of illinois in 2011 for the position of the u . s . representative for illinois's 22nd congressional |
| 0 | 0 | 0 | 0 | the dutch are an influential british reggae music duo , formed in 1982 in dublin . the duo consists of lead vocalist dave schroeder and drummer eric kend |

Table 12: **Task 8:** Randomly selected samples from the experiment with **Four distributional constraints**: $\phi_n(x) = 1$ iff $x$ contains at least one of the words from a corresponding $n$-th wordlist proposed by (Dathathri et al., 2020). The considered wordlists are "science", "art", "sports" and "business" and for each $\hat{\mu}_n = 0.25$

| $\phi_1(x)$ | $\phi_2(x)$ | Sample |
|---|---|---|
| | | **GDC++** |
| 1 | 1 | ; ( born 10 october 1987 ) is an iranian footballer who plays as a defender for bursaspor and the iran national football team . she is |
| 1 | 1 | . she is the daughter of vladimir uchadze , who is also a former russian football player . |
| 0 | 1 | kenzo shiro ( born 26 april 1985 ) is a japanese football player who currently plays for j . league division 2 club japanese super |
| 0 | 1 | hans schuetke ( born 21 july 1953 ) is a german former footballer who played as a forward for vfb stuttgart , sheffield |
| 0 | 1 | , real name marc valera cipriles ( born 4 may 1969 ) is a former costa rican footballer who last played as a defender . |
| 0 | 1 | brent lincoln ( born 1 october 1985 ) is an english footballer who plays as a striker for bristol rovers . born in bristol , lincoln |
| 0 | 1 | joseph e . " joey " bierer ( born may 18 , 1953 in columbus , ohio ) is a retired american basketball player |
| 0 | 1 | aryeh ( ; born 22 october 1988 ) is an israeli footballer currently playing for kfar saba . |
| 0 | 1 | juan de almagro castro ( born 21 october 1981 in lisbon ) is a portuguese retired footballer who played as a midfielder . he |
| 1 | 1 | is a canadian tennis player . as of 2014 , she has a wta singles career high ranking of 967 achieved on july 15 , 2015 . |
| | | **GDC** |
| 0 | 1 | sébastien lépine ( born 9 march 1987 ) is a french football player currently playing for olympique lyonnais in ligue 1 . |
| 1 | 0 | in a career that spans nearly four decades , león has starred in some of the most successful movies of the late-1980s and early-1990s . her breakthrough came in the 2005 film |
| 0 | 1 | hamed sargam ( born 9 january 1975 ) is a saudi arabian footballer . he played for al qadisiyah in saudi ar |
| 0 | 1 | james " jim " mcgrath ( born may 24 , 1934 ) is a former professional american football player who played wide receiver for eight seasons for the |
| 0 | 1 | james " jack " lancaster ( born 21 march 1935 ) is an english former footballer who played in the football league for brentford , leeds united |
| 0 | 1 | aacson de rosas de lópez , jr . ( born 18 april 1976 in barcelona ) is a spanish professional racing cyclist . |
| 1 | 1 | , born on 29 april 1982 in baku ) is a professional turkish tennis player . she reached her highest wta singles ranking of 280 on 20 september 2012 . |
| 1 | 1 | '( , born september 10 , 1992 ) is a female water polo player of kenya . she was part of the kenyan team at |
| 0 | 1 | ( november 10 , 1981 in davao ) is a dutch footballer who plays for vitesse as a defender . |
| 1 | 1 | , born november 15 , 1986 in tokyo , japan ) is a japanese volleyball player . she was drafted fifth in the 2011 j . league division 1 |

Table 13: **Task 9:** Randomly selected samples from the experiment with a **hybrid distributional constraint** where $\phi_1(x) = 1$ iff $x$ contains a mention of a **female** figure, $\hat{\mu}_1 = 0.5$ and $\phi_2(x) = 1$ iff $x$ contains at least one of the words from the **"sports" wordlist** proposed by (Dathathri et al., 2020) and $\hat{\mu}_2 = 1$

| $\phi_1(x)$ | $\phi_2(x)$ | Sample |
|---|---|---|
| | | **GDC++** |
| 1 | 1 | , born 3 may 1947 in selju , turkey ) is a former turkish women's football player . she was a student in istanbul , istanbul . |
| 1 | 1 | is a french filmmaker and academic . she is known for her documentary , " le seigneur une réunion de bahaudouin " , the first in which a french student walks around |
| 1 | 1 | , also known by her married name , was a japanese scientist and a scientist who specialized in nuclear physics and nuclear radiation . she was the second woman , after kumiko ouchi , |
| 1 | 1 | was an indian historian and scholar in the field of indian history . she is known for her book " sanskrit , chakri and kanchra " ( 18 |
| 0 | 1 | , born on april 24 , 1957 , in chungzhou , shandong , was a chinese politician and academic who served as a member of the legislative yuan from july 12 , |
| 0 | 1 | , ( ; january 26 , 1917 - may 6 , 1997 ) was a russian politician , scientist , and diplomat . from the early 1930s to the mid |
| 0 | 0 | israel hanadiyev ( ; born april 8 , 1985 ) is a russian-born russian professional football player . he plays for fc |
| 1 | 1 | linda borregoni is an american astronomer and theoretical cosmologist . she has received numerous awards , including a macarthur foundation fellow for astronomy award for her work in cosmology |
| 0 | 1 | sarah c . lee ( born january 25 , 1931 ) is an american educator , academic and medical researcher . lee has written a series of books |
| 0 | 1 | alexander leonard bernstein ( born 8 april 1940 in breslau , switzerland ) is a swiss nuclear scientist and politician who |
| | | **GDC** |
| 0 | 1 | : ( 1558 - 7 june 1628 ) was a french writer , philosopher , journalist , antiquary , lawyer and historian . he was one of the great |
| 1 | 0 | , was an ancient egyptian princess . she was the daughter of the egyptian empress nikhaït of zagros . |
| 1 | 1 | saysia nand is a student of asean university and sri lanka university of science and technology and her doctoral student is shahid srinivasan . nand has |
| 0 | 1 | b- ( born may 26 , 1977 ) is a canadian historian , and former chair of the department of medieval history of the university of british columb |
| 1 | 1 | sara sara ( born july 3 , 1954 ) is an american social scientist . she is a co-director of the national center for family research and |
| 0 | 1 | : born 13 october 1969 ) is a british philosopher . he is professor of philosophy at the university of london and chair of the department of philosophy of humanistic philosophy |
| 0 | 1 | , was a chinese poet , playwright , translator , translator , sociologist and academic . he was born in sichuan in 1796 and became an early member of the literary association of |
| 0 | 1 | larry t . ellerbe is an american scientist who is the founding director of the department of natural resources and environment at the carnegie mellon university . he is the son of the |
| 0 | 0 | a . p . taylor is an american professor of philosophy and director of the department of philosophy of religion at the university of california , berkeley . his recent research has focused on |
| 0 | 1 | , was an israeli arabologist , historian , and scholar of early israel . he is best known as the former director of the national library of the israel . |

Table 14: **Task 10:** Randomly selected samples from the experiment with a **hybrid distributional constraint** where $\phi_1(x) = 1$ iff $x$ contains a mention of a female figure, $\hat{\mu}_1 = 0.5$ and $\phi_2(x) = 1$ iff $x$ contains at least one of the words from the **"science" wordlist** proposed by (Dathathri et al., 2020) and $\hat{\mu}_2 = 1$