# OpenReview forum: "On Reinforcement Learning and Distribution Matching for Fine-Tuning Language Models with no Catastrophic Forgetting"
_NeurIPS.cc/2022/Conference — NeurIPS 2022 Accept_

### Official Review · Reviewer_6JLh · 2022-07-11

**Rating:** 7
**Confidence:** 4
**Soundness:** 3 good
**Presentation:** 3 good
**Contribution:** 3 good

**Summary:**

This paper studies controllable text generation with an emphasis on sequence level rewards and distribution matching. It first establishes a connection between reward maximization (RM) with KL control and distribution matching (DM) in a general sense. They show that there is an equivalent formulation of RM in DM terms and also DM can be connected to RM by thinking the importance sampling (IS) term as a reward. Using these connections, the authors highlight that a recent DM method, distributional policy gradient (DPG), suffers from the same high variance of REINFORCE estimators as the latter appears in the loss function of DPG. As is typical in RL, the authors propose using advantage estimations to reduce the variance of the DPG estimator and show that the intuitive expected reward (estimated as mean IS weights) gives unbiased estimates. The authors evaluate the proposed method, GDC++, on 10 controllable generation tasks including pointwise and distributional constraints using moment matching, KL-divergence, BLEU score metrics. When compared to other baselines, GDC++ shows lower divergence from the prior language model while also having competitive constraint matching. Using an advantage estimation is also shown to perform better and more stable results.

**Questions:**

I have several questions regarding my concerns above.

1. Could you clarify how your derivations are related to prior work on distribution matching and max entropy RL ?
2. Can you clarify the form of KL divergence used in the paper?
3. What is the relationship between the baseline used in the paper and that of Ziegler?
4. Could you clarify how Ziegler is trained, especially $\beta$ hyperparameter? Can you get better tradeoff by tuning this hyperparameter?
5. Can you discuss how these metrics translate into human satisfaction?

**Limitations:**

The authors addressed the limitations.

**Strengths And Weaknesses:**

The main strength of the paper is the connection between RM and DM to motivate using a baseline in DPG. There are also several weaknesses that needs clarification. I detail these below.

## Strength
1. The paper is well written and easy to follow.
2. The connection between RM and DM to motivate using a baseline in DPG and showing that the particular expected reward baseline is unbiased are interesting.
3. Experimental results show improvement compared to GDC and Ziegler et al.

## Weaknesses
1. While DM is not exactly the same as RM, I think there are previous work that incorporates KL-divergence between an online distribution and an offline distribution as a reward signal to train a policy using RL. Some of the more recent work are SAC$^1$, SMM$^2$ that induce a reward using a distribution matching and an older work *. While these don't necessarily utilize a KL-control, they study KL-divergence objective with energy based models or reward shaping.
2. The KL-divergence appears in different forms at different places. You define DPG in Line 105 and Algorithm 1 using $D_{KL}(p || \pi_\theta)$ while in Line 135 it is reverse.
3. Ziegler uses a reward shaping with a KL-control and the model is optimized via PPO. As PPO uses GAE, it also subtracts a baseline from this new reward. I think the relationship between this and the proposed baseline needs to be addressed.
4. It is also not clear how the $\beta$ is tuned in Ziegler baseline. By varying this hyperparameter, Ziegler can do better on $D_{KL}(\pi_\theta, a)$ while doing worse on $E_{\pi_theta} \phi(x)$, performing very similar to GDC++.
5. There is no human evaluation study in the paper and it is not clear how to judge the metrics proposed. For example, does higher KL from the language prior always means worse generation?

$^1$ Soft Actor-Critic: Off-Policy Maximum Entropy Deep Reinforcement Learning with a Stochastic Actor.
$^2$ Efficient Exploration via State Marginal Matching.
\* Reinforcement Learning by Probability Matching.

---

> ### Author Response · Authors · 2022-08-02
> **Response to Reviewer 6JLh, part 1**
>
> Thank you for your detailed and thoughtful reviewer. We are glad you found our paper well-written and interesting. Please find responses to your questions below.
>
> > Could you clarify how your derivations are related to prior work on distribution matching and max entropy RL?
>
> We explored connections to these lines of work but we had to move this discussion to Appendix B (lines 729-744) due to space limitations. We now pursued your suggestion by expanding it further (lines 745-790) to discuss relations to methods such as SQL (Haarnoja et al., 2017), SAC (Haarnoja et al., 2018), AWR (Peng et al., 2018) and SMM (Lee et al., 2019).
>
> In short, KL-control can be seen as a generalisation on MaxEnt RL to informed priors (in our case, the prior $a$ is a pretrained LM while MaxEnt RL assumes a uniform prior). Actually, both KL-control and MaxEnt RL can be seen as special cases of the control-as-inference framework (Levine, 2018). However, most practical algorithms in the MaxEnt RL family minimise KL from a target policy which changes throughout training; in contrast, DPG’s target distribution $p$ and KL-control implicit target distribution $p_z$ are defined at trajectory level and fixed throughout training. We will happily move parts of this discussion back to the main text if the paper is accepted (giving us one more page than we have for submission).
>
> > Can you clarify the form of KL divergence used in the paper?
>
> The difference in the order of arguments in KL is the major difference between DPG and KL-control, the two approaches we compare in the paper:
> 1. DPG *explicitly* minimises the *forward* KL from its target distribution $p$ (eq. 3): $\text{KL}(p, \pi_\theta)$,
> 2. KL-control *implicitly* (as we show) minimises the *reverse* KL from a certain distribution $p_z$ (eq. 10): $\text{KL}(\pi_\theta, p_z)$.
>
> > What is the relationship between the baseline used in the paper and that of Ziegler?
>
> Ziegler (2020) is implemented using PPO with a Generalised Advantage Estimator (GAE) based on a learned value function while the baseline we introduce here is tailored for the parametric reward in the distributional matching objective. Both baselines aim at reducing variance, yet for two different processes Reward Maximisation and Distribution Matching respectively. The two most important differences between these two baselines are:
> 1. Ziegler’s baseline is defined at token-level, i.e. for each token in a sequence. In contrast, DPG’s baseline is at the level of sequences seen as units.
> 2. Ziegler’s baseline is given by a value function estimating, for each token in a sequence, the return (in our case: sequence-level reward). The value function is implemented as a separate head on top of the LM. In contrast, DPG’s baseline is a constant estimated by a moving average of importance sampling estimates of $Z$ over the course of training.
>
> As we mention in the conclusion section, an interesting direction of future work would be to explore the  actor-critic version of DPG. It is important to note here that importing the standard RL form of value functions for the distributional matching objective and parametric rewards of DPG is not guaranteed to be correct. Hence, a non-trivial theoretical effort is needed to find the form of the value function. Such an algorithm might be more efficient as it will be capable of exploiting the sequential nature of $x$.
>
> > Could you clarify how Ziegler is trained, especially β hyperparameter? Can you get better tradeoff by tuning this hyperparameter?
>
> Ziegler’s $\beta$ is not constant, but undergoes updates during training. Our implementation closely follows the adaptive schedule described by Ziegler et al. (2020, p. 3). They vary $\beta$ to target a predefined value of $\text{KL}(\pi_\theta, a) = 6$ and update it $\beta$ throughout training according to equations provided in their paper.
>
> This results in unstable training as seen on learning curves, e.g. on Figure 3 or Figures 7-8 in the Appendix of our paper. One might consider that the need for an adaptive schedule for beta and resulting instability is a limitation of Ziegler et al. (2020). (In contrast, DPG does not require a beta hyperparameter.)
>
> Haarnoja et al. (2017). Reinforcement Learning with Deep Energy-Based Policies
>
> Haarnoja et al. (2018). Soft Actor-Critic: Off-Policy Maximum Entropy Deep Reinforcement Learning with a Stochastic Actor
>
> Lee et al. (2020). Efficient Exploration via State Marginal Matching
>
> Levine (2018). Reinforcement Learning and Control as Probabilistic Inference: Tutorial and Review
>
> Peng et al. (2019). Advantage-Weighted Regression: Simple and Scalable Off-Policy Reinforcement Learning
>
> Ziegler et al. (2020). Fine-Tuning Language Models from Human Preferences

---

> > ### Comment · Reviewer_6JLh · 2022-08-09
> > **Response to Authors**
> >
> > Thank you for the clarification, I will increase my score.
> >
> > Just one additional point. Could you please remove Lines 788-790 or add a proof? It is not clear to me if no MDP can formulate the problem where the SMM can be applied to recover your objective.

---

> > > ### Author Response · Authors · 2022-08-09
> > > **Response to Reviewer 6JLh**
> > >
> > > Thanks a lot! We agree that lines 788-90 were unsatisfactory, we just uploaded a revised version of the appendix with lines 788-90 removed.

---

> ### Author Response · Authors · 2022-08-02
> **Response to Reviewer 6JLh, part 2**
>
> > Can you discuss how these metrics translate into human satisfaction?
>
> Concerning the question whether “higher KL from the language prior $a$ always means worse generation”, if by “worse generation” one means “less fluent”, then in principle the answer is *no*. For example, with a pointwise constraint, if the model $\pi_\theta$ was concentrated on a single sentence both respecting the constraint and of high fluency (e.g. as measured through $a$), then the “quality” of the model would be high but $\text{KL}(\pi_\theta, a)$ would be larger than $\text{KL}(p, a)$, where $p$ is the optimal model from GDC’s perspective (which DPG tries to reach). However $\pi_\theta$ would have zero diversity!
>
> Thus the question of human evaluation is a delicate one, because it strongly depends on the chosen evaluation protocol, in particular the role of diversity. In relation to this aspect, we include two downstream diversity metrics: Distinct-$n$ score and Self-BLEU, inspired by existing work in the field of controllable language generation (see Zhang et al (2022) for a survey). We note, however, that (1) diversity may be difficult to gauge for human evaluators and (2) the notion of diversity itself is multifaceted. In the case of GDC/DPG, the target distribution $p$ does not only favour diversity in the sense of generating different texts respecting the constraints, but also in the sense of being representative of the underlying space of sentences generated by $a$ that satisfy the constraints. If one agrees that this is a reasonable approach, then the proper way to assess the model appears to be through formal KL divergence measures such as the ones we compute.
>
> Overall, we feel that the question of human evaluation of controllable generation is far from closed, but that properly addressing it would lead us too far away from the main thrust of the current paper. We welcome further discussion about that and could add a clarification in the final version or its appendix.
>
> Zhang et al. (2022). A Survey of Controllable Text Generation using Transformer-based Pre-trained Language Models.

---

### Official Review · Reviewer_wVhU · 2022-07-12

**Rating:** 6
**Confidence:** 3
**Soundness:** 3 good
**Presentation:** 2 fair
**Contribution:** 3 good

**Summary:**

This paper focuses on fine-tuning language models for controllable language generation tasks ( constraining topic, sentiment, gender distributions, etc). The authors derive the connection between distribution matching approaches and approaches that use KL-divergence penalty to prevent distribution shifts. By deriving a parametric reward baseline for Distributional Policy Gradients (DPG) and use it to extend the work by Khalifa et al., a distribution matching approach for nudging autoregressive generation distributions, this work achieves variance reduction and superior performance.

**Questions:**

Overall, I think the technical contributions are solid and novel, but I have concerns about how these contributions are framed and positioned. Can the authors address or clarify the concerns I raised above?

**Limitations:**

Limitations and potential negative societal impact are not discussed. Please add relevant pieces to the conclusion section.

**Strengths And Weaknesses:**

Strengths:
* The mathematical derivations for connecting policy gradients and distributional policy gradients (DPG),  connecting KL-control and distributional matching, DPG baseline are easy to follow. I appreciate the effort. These derivations provide good perspectives and should be valuable to the community.
* The main claim of introducing a baseline to reduce variance of DPG is well-justified and supported by evidence (Section 4).

Weaknesses:
* I think the contribution on making connection between policy gradients and distributional policy gradients is unnecessarily exaggerated, such as line 77: "So far, the connections between these two seemingly distinct paradigms have not been thoroughly explored". I think even the similarity in names already suggest that they are closely related, and they indeed aren't that different from each others. I think this claim actually makes the paper's logic difficult to follow for me. I would suggest focusing more on the contribution of introducing the parametric baseline, which should be the main product of this paper in my opinion.
* I'd like to see a consolidated definition of the control language generation task. Right now, they are scattered in intro, Sec. 4.1, Sec. 4.2. What's the context and desired outcomes of such tasks? How does DPG achieve the desired outcomes (now it's sort of described in Sec. 4.2)? Can the authors give some examples?

---

> ### Author Response · Authors · 2022-08-02
> **Response to Reviewer wVhU**
>
> Thank you for your constructive feedback and appreciating the merits of our work. We reply to your questions and concerns inline.
>
> > I have concerns about how these contributions are framed and positioned.
>
> Thank you for your suggestion to enhance the presentation of our contributions, we rephrased the contributions at the end of the introduction section to convey your suggestions.
>
> > a consolidated definition of the control language generation task.
>
> Controlled language generation tasks can be consolidated as follows. We are given a pre-trained language model $a(x)$ and a set of constraints, each defined as a binary feature $\phi_i(x)$ and a corresponding desired moment $\bar{\mu}_i\$, the percentage of LM samples satisfying the constraint. The goal is to obtain a fine-tuned language model $\pi_\theta(x)$ that satisfies the imposed constraints while deviating as minimally from the original language model $a(x)$.
>
> For example, in task 5 we follow (Dathathri et al. 2019 and Khalifa et al. 2021) and control an LM to produce solely positive sentiment generations. We utilise a binary sentiment classifier $\phi(x)$ and define its expected desired percentage in the output text $(\bar{\mur = 1$), i.e. the fine-tuned language model should produce positive sentiment samples 100% of the time.
>
> In task 8, we would like to balance generations between 4 topics (“science”, “sports”, “arts” and “business”). For this we build 4 different features $\phi_\text{science}(x), \phi_\text{art}(x), \phi_\text{sports}(x), \phi_\text{business}(x)$; each of them is a simple rule-based feature outputting 1 if the generation contains any word in a wordlist representing the corresponding topic. We set the desired expectation of each feature to be 0.25. i.e. the fine-tuned language model should produce on average 25% of the generations about science, and similarly for business, arts and sports respectively.
>
> Each of the methods should take an initial language model as input along with a set of predefined constraints. Each fine-tuned model can then be evaluated based on its constraint satisfaction rate and deviation from the original language model (catastrophic forgetting). This deviation is measured in terms of KL divergence as well as other linguistic qualities such as sentence-level and corpus-level diversity using Distinct-$n$ score and Self-BLEU.
>
> We rewrote sections 4.1 and 4.2 to reflect this consolidated view. Please check the updated manuscript. We are grateful for motivating us to enhance the readability of this section.
>
> > Limitations and potential negative societal impact are not discussed. Please add relevant pieces to the conclusion section.
>
> Thank you for your suggestion. We discussed the societal impacts of large language models (and how fine-tuning methods can be effective in alleviating those social risks but also subject to misuse) in Appendix A. We agree this is an important topic to include in the main text. We have updated the manuscript to add the relevant pieces to the conclusion section with a reference to the appendix for broader discussion. We have also included a new reference (Weidinger et al. 2021) that highlights the risks of “distributional stereotyping” and how Distributional Matching approaches can be suitable in reducing its harms.
>
> Overall, Thank you for your constructive feedback, we hope to have addressed your questions and concerns through our response and the updated version of our manuscript. We are happy to answer any further questions you have during the discussion period.
>
> Dathathri et al. (2020). Plug and play language models: A simple approach to controlled text generation
>
> Weidinger et al. (2021). Ethical and social risks of harm from Language Models

---

### Official Review · Reviewer_fBV8 · 2022-07-18

**Rating:** 7
**Confidence:** 4
**Soundness:** 4 excellent
**Presentation:** 4 excellent
**Contribution:** 3 good

**Summary:**

The paper elaborates on the idea of the DPG (Parshakova et al., 2019b) and GDC (Khalifa et al., 2021) paper, explicitly states that "KL-control", a specific reward maximization method, is also a distribution matching method with energy-based models. The paper also clearly explains the similarity and difference between distributional gradient descent and standard gradient descent. An advantage-based GDC algorithm is introduced in the paper as a standard technique in RL to reduce the variance of gradient estimates. Numerical experiments show the effectiveness of the proposed method.

**Questions:**

I don't have any critical questions or confusions about the paper.

**Limitations:**

Limitations and future directions are well discussed at the end of the paper.

**Strengths And Weaknesses:**

DPG framework could be a powerful tool for fine-tuning LMs to maximize some non-differentiable objectives using RL/EBM, but also prevent the catastrophic forgetting problem. The idea is neat with mostly straightforward derivation. The paper clarifies the key concepts and connections to existing training paradigms. Readers that are perplexed about fine-tuning LMs with reinforcement learning would greatly benefit from this paper.

---

> ### Author Response · Authors · 2022-08-02
> **Response to Reviewer fBV8**
>
> Thank you for feedback and the overall positive assessment of our submission, namely the theoretical connections between DPG and standard policy gradient and the connection between KL-control and DM methods, the introduction of a baseline to the GDC algorithm and its empirical effectiveness for reducing variance during the finetuning process.
>
> Indeed one of our objectives for writing this manuscript was to provide a self-contained, intuitive guide elaborating the similarities and differences between existing training paradigms for finetuning LMs using Reinforcement Learning. Some of these connections (e.g. the connections between KL-control and Distribution Matching) were initially unclear to us and sparked hours of discussion. We thought sharing those insights with the community could benefit future work in that direction to be more theoretically grounded. We are happy to see such contributions appreciated by you. Thank you, and we are available to reply to any further questions that you have.

---

### Author Response · Authors · 2022-08-02
**Revised version of the paper**

We are grateful to all three reviewers for their positive assessment and thoughtful comments. Based on that feedback, we revised both the main text and the appendix. In the main text, we rewrote parts of Section 4 to include a consolidated definition of the controllable language generation task and Section 5 to include discussion of the ethical implications of our work. We also extended the discussion of related work in Appendix B to cover approaches such as MaxEnt RL and state marginal matching in greater depth. If the paper is accepted, we will use the additional tenth page to further expand on these points in the main text.

---

### Meta-Review · Area_Chair_JWJG · 2022-08-28

**Recommendation:** Accept
**Confidence:** Certain

**Metareview:**

All reviewers consistently agree that this paper provides valuable contribution in identifying the connection between different RL training paradigms for language models with clear derivations. In addition, it also develops an improved method by incorporating a baseline into DPG.

**Award:**

No

---

### Decision · Program_Chairs · 2022-09-14

Accept